# Unraveling the Potential of ALK-Targeted Therapies in Non-Small Cell Lung Cancer: Comprehensive Insights and Future Directions

**DOI:** 10.3390/biomedicines12020297

**Published:** 2024-01-27

**Authors:** Hannaneh Parvaresh, Ghazaal Roozitalab, Fatemeh Golandam, Payam Behzadi, Parham Jabbarzadeh Kaboli

**Affiliations:** 1Department of Biology, Faculty of Science, Ferdowsi University of Mashhad, Mashhad 9177948974, Iran; 2Division of Cancer Discovery Network, Dr. Parham Academy, Taichung 40602, Taiwan; ghazaal.rzt@gmail.com (G.R.);; 3Noncommunicable Diseases Research Center, Fasa University of Medical Sciences, Fasa 7461686688, Iran; 4Department of Pharmacy, Mashhad University of Medical Science, Mashhad 9177948974, Iran; 5Department of Microbiology, Shahr-e-Qods Branch, Islamic Azad University, Tehran 37541-374, Iran; behzadipayam@yahoo.com; 6Graduate Institute of Biomedical Sciences, Institute of Biochemistry and Molecular Biology, China Medical University, Taichung 407, Taiwan

**Keywords:** NSCLC, ALK inhibitors, lung cancer, targeted therapy, precision medicine

## Abstract

**Background and Objective:** This review comprehensively explores the intricate landscape of anaplastic lymphoma kinase (ALK), focusing specifically on its pivotal role in non-small cell lung cancer (NSCLC). Tracing ALK’s discovery, from its fusion with nucleolar phosphoprotein (NPM)-1 in anaplastic large cell non-Hodgkin’s lymphoma (ALCL) in 1994, the review elucidates the subsequent impact of ALK gene alterations in various malignancies, including inflammatory myofibroblastoma and NSCLC. Approximately 3–5% of NSCLC patients exhibit complex ALK rearrangements, leading to the approval of six ALK-tyrosine kinase inhibitors (TKIs) by 2022, revolutionizing the treatment landscape for advanced metastatic ALK + NSCLC. Notably, second-generation TKIs such as alectinib, ceritinib, and brigatinib have emerged to address resistance issues initially associated with the pioneer ALK-TKI, crizotinib. **Methods:** To ensure comprehensiveness, we extensively reviewed clinical trials on ALK inhibitors for NSCLC by 2023. Additionally, we systematically searched PubMed, prioritizing studies where the terms “ALK” AND “non-small cell lung cancer” AND/OR “NSCLC” featured prominently in the titles. This approach aimed to encompass a spectrum of relevant research studies, ensuring our review incorporates the latest and most pertinent information on innovative and alternative therapeutics for ALK + NSCLC. **Key Content and Findings:** Beyond exploring the intricate details of ALK structure and signaling, the review explores the convergence of ALK-targeted therapy and immunotherapy, investigating the potential of immune checkpoint inhibitors in ALK-altered NSCLC tumors. Despite encouraging preclinical data, challenges observed in trials assessing combinations such as nivolumab-crizotinib, mainly due to severe hepatic toxicity, emphasize the necessity for cautious exploration of these novel approaches. Additionally, the review explores innovative directions such as ALK molecular diagnostics, ALK vaccines, and biosensors, shedding light on their promising potential within ALK-driven cancers. **Conclusions:** This comprehensive analysis covers molecular mechanisms, therapeutic strategies, and immune interactions associated with ALK-rearranged NSCLC. As a pivotal resource, the review guides future research and therapeutic interventions in ALK-targeted therapy for NSCLC.

## 1. Introduction

Lung cancer comprises non-small cell lung cancer (NSCLC) (81% of cases) and small cell lung cancer (SCLC) (14% of cases). In the US, NSCLC dominates and is projected to affect around 238,340 adults by 2023, resulting in 127,070 deaths. Globally, 2,206,771 people were diagnosed with lung cancer in 2020, encompassing both NSCLC and SCLC cases [1,2].

Anaplastic lymphoma kinase (ALK), belonging to the insulin receptor superfamily, plays a significant role in various cancers. The *ALK* gene is in the 2p23.2-p23.1 chromosomal region, encoding a protein of 26 exons and 1620 amino acid residues. In 1994, a groundbreaking discovery revealed the fusion between nucleolar phosphoprotein (NPM)-1 and ALK in anaplastic large cell non-Hodgkin’s lymphoma (ALCL), emphasizing ALK’s importance. Understanding the t(2;5)(p23;q35) chromosomal translocation in ALCL led to identifying ALK and the resulting NPM-ALK oncogenic protein [3,4]. On the other hand, different changes in the *ALK* gene—such as alternative splicing, amplification, and mutations—are linked to various tumors, such as inflammatory myofibroblastoma and NSCLC [4,5,6,7]. In addition, *ALK* gene rearrangements impact immune systems, affecting T-cell activation, cytokine secretion, and immune evasion within tumors [8]. This diversity makes ALK gene variations promising targets for cancer therapies [9,10]. 

In 2007, the *ALK* gene rearrangement in NSCLC patients revealed the initial fusion between echinoderm microtubule-associated protein-like 4 (EML4) and ALK in lung cancer [11,12]. Around 3–5% of NSCLC patients exhibit ALK rearrangement, often associated with a non-smoking history, younger age, and adenocarcinoma histology [13,14]. Six ALK-tyrosine kinase inhibitors (TKIs) (crizotinib, ceritinib, alectinib, brigatinib, ensartinib, and lorlatinib) received approval by 2022 for advanced metastatic ALK-rearranged (ALK+) NSCLC treatment. Among these, alectinib, brigatinib, and lorlatinib are recommended for advanced ALK+ NSCLC in the United States [15]. Crizotinib was the first approved ALK-TKI inhibitor for treating ALK-rearranged NSCLCs. Studies indicated its significant efficacy compared to standard chemotherapy [16].

Crizotinib has demonstrated significant efficacy in reducing tumor size by approximately 50% to 60% in patients with ALK protein alterations, even among those previously treated with chemotherapy. Typical side effects encompass nausea, vomiting, diarrhea, constipation, bloating, fatigue, edema, and eye issues. At the same time, more severe outcomes may involve reduced leukocyte count and detected changes in the lungs and heart. Moreover, clinical investigations highlight the robust efficacy of a co-targeting approach, combining epidermal growth factor (EGF) receptor (EGFR)-TKIs with crizotinib as targeted therapies, especially in metastatic NSCLC [17]. Furthermore, combining crizotinib with immune checkpoint inhibitors (ICIs) targeting programmed cell death protein 1 (PD-1), programmed cell death-1 ligand-1 (PD-L1), and cytotoxic T-lymphocyte-associated protein 4 (CTLA-4) show potential for NSCLC treatment. Still, their efficacy in oncogenic mutated proteins such as EGFR or ALK remains uncertain. ALK alterations have been linked to increased immune checkpoint expression, raising questions about the effectiveness of immunotherapy alone or combined with targeted therapies in this subset of patients. However, trials evaluating immunotherapy in NSCLC often need more representation of ALK-rearranged patients, limiting robust conclusions about their clinical benefit in this population [18,19].

However, second-generation ALK-TKIs have demonstrated superior clinical activity in terms of median progression-free survival (PFS), objective response rate (ORR), intracranial disease control, and duration of response when compared to crizotinib. The second-generation ALK-TKIs are the gold-standard first-line treatment for ALK-rearranged metastatic NSCLC. Among these options, alectinib is considered to have the most favorable profile of clinical activity and safety, making it a preferred choice for upfront therapy. Ongoing trials and biomarker analyses will provide further insights into the optimal treatment approach [20]. Additionally, second-generation ALK-TKIs (alectinib, ceritinib, brigatinib) were developed to combat resistance emerging with crizotinib. Initially, these TKIs showed promising effectiveness, validated in several phase 3 trials as the primary treatment for newly diagnosed ALK+ NSCLC [21,22,23,24,25]. However, resistance mechanisms may be ALK-dependent or ALK-independent, involving bypass signaling pathways and histological transformation, which could impact subsequent therapy decisions [26,27].

The current review comprehensively analyzes ALK-rearranged NSCLC, delving into mechanisms and updated data concerning ALK-targeted therapy and immunotherapy. Its scope includes ALK’s structural biology, tissue-specific functions, and diverse roles in ALK-targeted therapy. Additionally, it thoroughly explores the signaling pathways activated by ALK fusion proteins and mutations, addressing challenges in ALK-targeted therapy resistance and proposing innovative strategies, notably combination therapies. Ultimately, this review is a valuable resource, offering insights for future research and guiding therapeutic interventions in the domain of ALK-targeted therapy for NSCLC.

## 2. The ALK Structural Biology

### 2.1. ALK Extracellular Side

The ALK extracellular domain (ECD) consists of distinct segments believed to serve specific roles such as binding ligands, interacting with potential co-receptors and secreted regulatory proteins, and facilitating dimerization. These functions could trigger structural changes initiating activation within the intracellular protein tyrosine kinase (PTK) domain. The ECD of ALK stands out among receptor tyrosine kinases (RTKs) due to its distinct glycine-rich section. At the same time, ALK includes an additional low-density lipoprotein receptor class A (LDLa) and two meprin, A5 protein, and receptor protein tyrosine phosphatase mu (MAM) domains (Figure 1). Pleiotrophin (PTN) and midkine (MK) are recognized as triggers for mammalian ALK, playing pivotal roles in neural development, survival, and tumorigenesis [28]. These growth factors, binding to heparin, can activate various receptors, including receptor protein tyrosine phosphatase-β (RPTPβ), RPTPζ, N-syndecan, low-density lipoprotein receptor-related protein (LRP), and integrins. PTN can specifically engage RPTPβ and RPTPζ phosphatases to initiate ALK signaling. However, the actual activation of ALK by PTN and MK remains contentious, with conflicting reports among studies. This debate contrasts with findings in non-vertebrate models such as Drosophila melanogaster and Caenorhabditis elegans, underscoring the ongoing uncertainty about the natural ligand for mammalian ALK [3].

The structure of ALK deviates from the typical architecture due to its unique domain composition. One distinctive element is the glycine-rich domain (GRD) near the membrane. This GRD, which contains a cysteine-rich area resembling the fold of EGF, is an unconventional and less defined region. Interestingly, this peculiar GRD is a shared feature between ALK and LTK. Despite its high glycine content, often associated with structural disorders, the GRD alone can drive receptor activity regulated by its ligands. Vertebrate ALKs’ ligands, known as ALKALs (ALK and LTK Activating Ligands), include Fam150A (AUGβ) and Fam150B (AUGα). These ligands, consisting of approximately 100 amino acids, contain a highly conserved domain termed the ALKAL domain, which stimulates ALK activity [29,30,31]. A recent crystallographic study of the GRD reveals a ß-sandwich structure with N- and C-termini positioned close to each other, resembling a portion of the tumor necrosis factor-α (TNF-α) domain. Despite this similarity, the ß-sandwich structure of ALK differs from TNF-α. The loop in the C-terminal contributes to a fold, creating a binding epitope [29]. The crystal structure of ALK demonstrates that two ALKALs bind to the dimeric complex of two ALKs, stabilizing the complex. The ALK GRD comprises short α-helices, ß-sheets, and glycine helices, while the ALKAL domain forms a disulfide helical hairpin [29,32,33,34]. The interaction between ALK and ALKAL initially occurs through the ALKAL domain and the TNF-α-like region within the GRD, attributed to the high positive charge on the ALKAL domain surface, which supports ALK protein activation (Figure 1) [29].

### 2.2. ALK Intracellular Side

The activation loop (A-loop), a pivotal segment governing access to the active site, commences with a conserved Asp-Phe-Gly (DFG) sequence, significantly regulating ALK’s active and inactive states. ALK boasts two distinct hydrophobic spines named the “regulatory” and “catalytic”, contributing to vital allosteric control within and between the lobes. These spines, housing conserved hydrophobic amino acids, facilitate the transition between active and inactive states. The ALK regulatory spine, encompassing I1171, C1182, H1247, F1271, and D1311, assembles during kinase activation and disengages during inactivation [35]. Researchers also explored the structure of the unphosphorylated human ALK kinase domain in tandem with ATP-competitive ligands such as PHA-E429 and NVP-TAE684. This analysis provided invaluable insights into the distinct attributes of the ALK active site, aiding in the quest for selective ALK inhibitors. Specifically, the ALK-PTK-PHA-E429 structure uncovered a potential regulatory mechanism, linking a brief helical segment after the DFG motif to a two-stranded beta-sheet at the N-terminal. The ALK structure begins with an initial 13-residue segment featuring two β-strands (β1′ and β2′) before the bilobal protein kinase fold. This configuration encompasses an N-terminal lobe housing a core five-stranded β-sheet and an α-helix.

In contrast, the C-terminal lobe is primarily α-helical and accommodates the critical activation loop pivotal for enzyme activation. The gaps in the ALK structures, particularly in the complexes with PHA-E429 and TAE684, indicate regions of structural disorder within the protein. Comparative analysis with other kinase structures highlighted discrepancies in lobe closure and αC helix positioning, suggesting potential inactivity in the ALK-PTK structure due to these structural deviations (Figure 1).

A crucial hydrogen bond between specific residues implies an active kinase state. The study also underscored variations in hydrogen bond formations and structural conflicts within the ALK-PTK structures, influencing the initiation of a specific structural element termed β9 and affecting the formation of the substrate binding site. However, confirming these observations remains challenging due to the absence of an apoenzyme ALK structure, which would depict the natural state devoid of inhibitors. Nonetheless, consistent features observed in most ALK structures, such as the DFG helix and interactions involving specific residues, hint at a potential regulatory role in enzyme function [32].

Furthermore, ALK exhibits a distinctive autophosphorylation motif, Y**XXX**YY (Y**RAS**YY), within the A-loop. In instances of ALK fusions, the tyrosine at position Y1278 is the primary site for phosphorylation within this sequence. Notably, an inhibitory structural feature within the ALK kinase domain involves a short α-helix in the A-loop closely associated with the αC-helix. At the same time, a β-turn motif containing C1097 obstructs the region for substrate binding. This arrangement prevents Y1278 from being accessible for phosphorylation as it forms a bond with C1097 within the amino-terminal β-sheet [36,37,38]. These insights suggest that the initial activation of ALK could potentially involve the regulation of Y1278 phosphorylation, thereby freeing ALK from inactive structural constraints (Figure 2) [39].

The ALK protein’s structural intricacies and regulatory mechanisms are governed by specific amino acid residues within its sequence. Residues within the range of 1095–1401 display intermittent gaps due to structural disorder, encapsulating crucial segments such as the glycine-rich loop (1123–1128) and the activation loop (1271–1288), essential for ALK’s functional modulation. Among these residues, 1150 (K1150) and 1167 (E1167) stand out for their involvement in pivotal hydrogen bond formations, contributing to ALK’s enzymatic activity. Residue 1245 (F1245) notably interacts, potentially impeding the initiation of a critical structural element termed β9. Additionally, residues 1274 (A1274) and 1278 (Y1278) are significant: A1274 engages in clashes with F1245, impacting structural conformation, while Y1278 marks the conclusion of the DFG helix, influencing the formation of the substrate binding site. These residues within the ALK protein sequence intricately contribute to its structural stability, functional regulation, and inhibitor interactions, which are crucial for understanding its biological significance and potential therapeutic targeting [32].

### 2.3. Crizotinib and ALK Inhibition versus c-MET

Crizotinib, a drug with diverse effects on various kinases, demonstrates different inhibitory patterns in enzyme and cellular assays. While it affects multiple kinases in enzymatic tests, its cellular actions show potent inhibition, specifically on mesenchymal-epithelial transition factor (c-MET) and ALK (Figure 2). This selectivity is linked to distinct binding sites shaped by unphosphorylated c-MET’s unique conformation. However, in the ALK-PTK, crizotinib shows similarities in binding to c-MET but lacks a crucial interaction in ALK, possibly explaining its weaker potency against ALK. Critical interactions with specific amino acids (M1211 in c-MET) play a vital role in maintaining crizotinib’s inhibitory effect and are found similarly in RON (a c-MET-related receptor) and ALK [33]. This drug demonstrates promising potential in inhibiting c-MET and ALK phosphorylation, curbing tumor cell growth, exhibiting antiangiogenic properties, and inducing apoptosis in specific cancer cells. Eventually, preclinical and animal studies support its efficacy against cancers harboring ALK mutations, indicating its potential as a therapeutic agent. Clinical trials have shown remarkable effectiveness in several cancers, particularly in NSCLC and other tumors carrying fusion ALK genes or amplified c-MET genes [41]. 

Overall, crizotinib presents itself as a promising targeted therapy across diverse cancer types by selectively inhibiting c-MET and ALK, leading to crizotinib’s accelerated approval by the US Food and Drug Administration (FDA) on 26 August 2011, for treating ALK+ locally advanced or metastatic NSCLC. The approval relied on two single-arm trials, showcasing ORRs of 50% and 61%, along with median response durations lasting 42 and 48 weeks [42].

Furthermore, the co-crystal structure of the ALK kinase domain complexed with crizotinib has a binding configuration like c-MET. However, unlike c-MET, the interaction involving tyrosine π–π stacking is absent in the ALK co-crystal structure, potentially contributing to a slight decrease in potency against ALK compared to c-MET. Examining wild-type and L1196M ALK co-crystals aimed to delineate binding interactions and protein conformations [34]. The binding mode of crizotinib in both proteins exhibited remarkable similarity. Specific hydrogen bonds were formed between crizotinib and hinge residues M1199 and E1197 in wild-type and L1196M structures.

Additionally, interactions with the gatekeeper residue and other critical elements of the inhibitor were consistent across both structures, maintaining similar distances and positions. Although the overall protein conformations between the two crystal structures were broadly comparable, a notable difference was observed in the gatekeeper residue (L1196 to M1196). Further optimization was needed for the pyrazolopiperidine tail of crizotinib to enhance its potency against ALK and its ADME (absorption, distribution, metabolism, and excretion) properties. The pyrazole portion of the tail group occupied a lipophilic pocket near the solvent-exposed area. In contrast, the piperidinyl group extended toward the solvent, resulting in low permeability and high efflux. Strategies aimed at improving both the 2,6-dichloro-3-fluorophenyl head and the pyrazolopiperidine tail of crizotinib were explored to enhance its effectiveness against ALK [34].

In addition, ALK possesses multiple LC3-interacting region (LIR) motifs across different domains, hinting at a direct connection between ALK and autophagy. This suggests a complex interplay between inhibiting ALK kinase activity and activating autophagy, potentially complicating targeted therapies for NSCLC and other conditions. Understanding the dual roles of autophagy in cancer—serving as both an immune response facilitator and a tumor growth promoter—underscores the need to categorize ALK + NSCLC based on hepatocyte growth factor (HGF)/c-MET signaling or autophagy-related subtypes to guide treatment decisions for optimal patient outcomes (Figure 3) [43].

## 3. ALK Cleavage and Modifications

ALK protein manifests diverse sizes and variations, encompassing an 8.0 kb message detected in rhabdomyosarcoma, small intestine, and brain, alongside transcripts of around 6.5 kb, presumed as typical cDNA. Additionally, various ALK messages—approximately 6.0 kb in the human testis, placenta, and fetal liver, and a distinct 4.4 kb transcript found solely in the testis—highlight tissue-specific isoforms likely arising from alternative transcriptional start sites or polyadenylation signals. These distinct sizes potentially signify tissue-specific functions generated via alternative splicing, leading to diverse receptor forms with varying ligand-binding capabilities and biological activities. Investigations have striven to unravel the precise roles of these diverse ALK isoforms in specific tissues, shedding light on their significance in mammalian development and growth, particularly in neural signaling pathways and development [44].

A study delved into the expression of ALK in a specific subset of neurons associated with nociception and explored factors influencing its cleavage, shedding light on the potential roles of ALK in sensory neuron development and pain perception [45]. The study identified that 73% of these sensory neurons expressed ALK, with a significant portion also expressing markers for nociception. This suggests that ALK might be a marker for neurons sensing pain. Additionally, the study explored the impact of glial cells on ALK metabolism. It observed that Schwann cells release a factor that inhibits the cleavage of the ALK receptor into its two forms. The study proposes two hypotheses: one suggests direct binding of a factor to ALK, and another suggests inhibition of proteases involved in ALK cleavage. However, the experiments did not definitively identify the factor or clarify its mechanism [45].

The *ALK* gene has emerged as a significant player in neuroblastoma development, making it an attractive target for therapeutic interventions. Studies identified specific mutations at F1174 and R1275 in neuroblastoma tumor cells that activate ALK, establishing its role in the disease. Researchers clarified distinct behaviors between the standard and mutated forms of the ALK receptor. They identified that the altered ALK receptors are primarily inside the cell, notably in the reticulum/Golgi structures. This internal retention was particularly noticeable in the F1174L mutation compared to the R1275Q variation [46].

Treatments inhibiting ALK kinase activity resulted in the translocation of mutated receptors to the cell membrane. This sheds light on potential therapeutic avenues, suggesting that targeting ALK with kinase inhibitors or specific antibodies could hold promise in neuroblastoma treatment, especially considering the possibility of these antibodies inducing receptor internalization and downregulation. These findings open avenues for therapeutic approaches targeting both the wild-type and mutated ALK receptors in neuroblastoma treatment, offering potential complementary strategies to kinase inhibitors [46]. Furthermore, another study highlighted that different mutations in ALK could result in varying oncogenic potentials, with the ALK F1174L mutation exhibiting heightened activity. Importantly, it was found that the mutated receptor, especially the ALK F1174L variant, had altered trafficking patterns, predominantly retained inside the cell. Remarkably, treatment with specific inhibitors restored normal trafficking of the mutated receptor, suggesting a potential therapeutic approach. Additionally, the study unveiled complex mechanisms of ALK degradation, contingent upon its cellular location, offering insights into potential strategies to inhibit neuroblastoma proliferation by targeting these degradation pathways. Ultimately, this research underscores the complexity of ALK behavior and its implications for targeted therapies in neuroblastoma treatment [47].

Furthermore, a recent study introduced a potentially groundbreaking therapeutic approach targeting ALK using a peptide derived from neuronal growth regulator 1 (Negr1) [48]. Negr1 has been linked to regulating various RTKs, and the researchers observed that acute treatment with soluble Negr1 reduced ALK protein levels, suggesting it prompts ALK protein degradation [49]. The study proposed that the Negr1-derived peptide might influence ALK levels and downstream signaling pathways impacting cell proliferation. The peptides derived from Negr1 may interact differently with ALK compared to the full-length protein, potentially leading to ALK degradation. This could be a promising strategy as ALK activation is known to drive neuroblastoma growth, and therapies targeting ALK have shown efficacy but are prone to resistance and adverse effects. The Negr1-derived peptide demonstrated the ability to degrade ALK and slow tumor growth both in vitro and in vivo, presenting a promising avenue for treating aggressive neuroblastoma resistant to current ALK inhibitors [48].

On the flip side, although cleaving ALK’s intracellular domain may help ALK-targeted therapy, cleavage of its extracellular side fosters ALK-related tumor formation and the movement of cells in neuroblastoma. Another potential regulatory process involves the proteolytic breakdown of the full-length ALK receptor ECD [50], releasing an ECD fragment approximately 80 kDa in size alongside a significantly tyrosine-phosphorylated 140 kDa truncated receptor. The precise physiological significance and the molecular mechanisms driving this cleavage event remain ambiguous; it is uncertain whether cleaved ALK might be more stable or active than intact ALK and whether this cleavage plays a role in ligand-mediated activation [39]. 

Researchers noted that blocking this cleavage in neuroblastoma cells reduced migration and invasion. Intriguingly, introducing the cleavable form of ALK in cells with minimal ALK expression significantly boosted their migration, whereas mutations preventing cleavage did not have the same impact [51]. This indicates the critical role of this cleavage process in driving cell movement, supported by changes in gene activity linked to cell motion. Grasping this process’s developmental role is vital, as abnormal expression in neuroblastoma cells might heighten tumor spread. The study suggests this cleavage could affect a protein called β-catenin, regulating cell motion. When ALK undergoes cleavage at the N654-L655 site, it might release β-catenin, enabling its movement into the cell nucleus to activate genes related to cell motion. Conversely, obstructing matrix metallopeptidase 9 (MMP-9) could impede ALK cleavage, reducing migration and invasion of neuroblastoma cells, hinting at a promising therapeutic approach [51]. The study also explores the impact of ALK cleavage in other cancers where ALK is present and whether blocking this process could aid in devising new treatment approaches (Figure 4).

On the other hand, N-linked glycosylation impacts ALK function in neuroblastoma cells. However, it is essential to note that the N654 cleavage site targeted by MMP-9 activates β-catenin signaling. A substantial decrease in the binding of β-catenin to the truncated membrane-bound ALK 655-1604 receptor indicated that the cleavage of the ECD releases β-catenin from ALK, allowing its transportation to the nucleus [51]. Nonetheless, previous findings showed that N-linked glycosylations impact ALK [52] and other RTKs [53]. Researchers detected decreased ALK phosphorylation, specifically in neuroblastoma cells that depend on ALK for survival, when employing tunicamycin, a substance renowned for broadly disrupting glycosylation. Interestingly, this inhibition only affected cell proliferation and survival in ALK+ neuroblastoma cells, suggesting a potential therapeutic strategy. While tunicamycin broadly affects glycoproteins, its impact seemed selective for ALK-dependent cells. However, as tunicamycin might not be suitable for clinical use, the study’s findings emphasize a proof-of-concept, prompting further exploration into alternative approaches targeting N-linked glycosylation as a potential strategy for ALK-dependent neuroblastoma treatment [52]. 

Comprehensive research has delineated intricate molecular pathways involving diverse ALK isoforms, mutations, and regulatory mechanisms, particularly emphasizing their implications in neuroblastoma development and potential therapeutic strategies. Nevertheless, exploring ALK’s function, glycosylation impact, and cleavage mechanisms in NSCLC remains an uncharted territory. Therefore, future research should concentrate on elucidating the role of ALK in NSCLC, investigating how its glycosylation and cleavage intricately contribute to treatment resistance, which is pivotal for advancing effective therapeutic interventions in NSCLC patients relying on ALK-targeted therapy [54].

## 4. ALK Signaling and TKI Resistance

The majority of cases in NSCLC include a subset of two to seven percent of patients exhibiting gene rearrangements of the *ALK* gene or chromosomal fusions of ALK with echinoderm microtubule-associated protein-like 4 (EML4) [55,56]. Using ALK-TKIs has significantly improved the outcomes for NSCLC patients with these specific genetic abnormalities. Nevertheless, emerging evidence underscores the clinical challenge of primary or secondary resistance to ALK inhibitors during treatment, necessitating a shift to second- or third-generation ALK-TKIs and the meticulous monitoring of NSCLC patients on ALK-TKIs through repeated molecular testing [57]. The latest generation of ALK-TKIs offers benefits for most individuals with EML4-ALK fusions. However, resistance to ALK inhibitors can emerge due to point mutations within the kinase domain of the EML4-ALK fusion, such as G1202R, resulting in a reduction in the effectiveness of the inhibitors [58].

While several ALK inhibitors, such as crizotinib, alectinib, and ceritinib, have been utilized clinically for ALK+ NSCLC treatment, resistance commonly develops against these inhibitors. The mutated forms of ALK, along with ALK fusion proteins such as NPM-ALK, can activate various signaling pathways that contribute to cell transformation and the maintenance of a cancerous state. This persistent activation triggers the recruitment of several adaptors, initiating multiple signaling pathways. Mutated ALK and ALK chimeras induce mitogenic signaling, predominantly through the RAS/mitogen-activated protein (MAP) kinase pathway, facilitated by the direct binding of IRS1, SHC, and SRC to specific tyrosine residues within ALK’s intracytoplasmic segment. The SHP2/growth factor receptor-bound protein 2 (GRB2) complex interaction with p130Cas alters cytoskeletal organization. Activation of the phosphatidylinositol 3 kinase (PI3K) pathway by ALK results in a significant anti-apoptotic signal, mainly mediated by pAKT1/2 and its downstream molecules that inhibit BAD and FOXO3a-mediated transcription, while regulating cell cycle progression. Additionally, phospholipase C (PLC)-γ, directly binding to activated ALK, generates diacylglycerol and IP3, activating PKC and mobilizing calcium stores from the endoplasmic reticulum [59,60,61,62]. The Janus kinase (JAK)/Signal transducer and activator of transcription (STAT)-3 pathway activated by ALK provides crucial survival signals and regulates cellular metabolism via the mitochondrial oxidation chain. STAT3 activation, directly or through JAK, leads to a unique gene expression profile distinguishing ALCL from other T-cell neoplasms. Its downstream effectors include BCL2 family members (BCL2, BCL-XL, MCL-1), anti-apoptotic proteins such as survivin, and multiple transcription factors such as C/EBPβ. ALK fusion proteins also stimulate the upregulation of CD30 via RAS and AP-1 transcription factors, providing anti-apoptotic signals through TNF receptor-associated factor 2 (TRAF2) (Figure 5) [63]. 

Further, in neuroendocrine prostate cancer (NEPC), a severe form of prostate cancer, a mutation (ALK F1174C) in the *ALK* gene responded well to alectinib. An experimental model combining ALK F1174C and N-Myc led to aggressive NEPC, mirroring poor outcomes seen in human datasets. This combination also activated the wnt/β-catenin pathway [64]; however, as mentioned earlier, ALK cleavage by MMP-9 in neuroblastoma results in β-catenin release from ALK [51]. However, inhibiting ALK suppressed this pathway, hindering NEPC and neuroblastoma growth in lab experiments and live models. Combining ALK and Wnt inhibitors showed potential against NEPC and neuroblastoma, underscoring ALK’s significance and proposing a therapeutic strategy targeting both ALK and Wnt pathways in ALK-related tumors, linking insights between NEPC and neuroblastoma [64]. However, a subgroup of lung cancers relies on the ALK for survival, and treatment with the ALK inhibitor crizotinib initially yields remarkable tumor responses. Long-term effectiveness is limited due to emerging drug resistance.

Further investigation revealed that ALK controls MYC’s transcriptional expression and activates c-MYC’s regulation of target genes in NSCLC. Silencing MYC, either through RNAi or small molecules, sensitizes ALK+ cells to crizotinib. These findings illuminate a dual oncogenic mechanism whereby ALK stimulates the MYC signaling axis, suggesting that targeting MYC could potentially prevent or overcome crizotinib resistance [65]. 

In addition, findings suggested that targeting Src signaling could be a promising therapeutic strategy for ALK+ NSCLC cases that have developed resistance to ALK-TKIs. Researchers discovered that Src signaling is a key resistance mechanism to alectinib, and combining ALK and Src inhibitors effectively halted the growth of ALK-TKI-resistant cells. Further, blocking Src in alectinib-resistant cells effectively countered the activation of phospho-receptor tyrosine kinases and downstream PI3K/AKT signaling. This combined inhibition of ALK and Src also displayed effectiveness against other ALK+ NSCLC cell lines resistant to ceritinib or lorlatinib [66]. In addition, another group of researchers established ALK+ lung cancer cell lines resistant to ceritinib. Hence, treatment with ceritinib significantly increased Src activity. The silencing of Src alone using siRNA effectively restored sensitivity to ceritinib in ALK+ cells. Moreover, inhibiting Src with saracatinib was effective in ALK-resistant cancer cells. Therefore, ceritinib’s inhibition of ALK may trigger an upsurge in Src signaling, and saracatinib could potentially serve as a therapeutic agent for treating lung cancer patients resistant to ALK inhibitors [67].

On the other hand, in NSCLC, where ALK genes are rearranged, resistance to ALK-TKIs remains a challenge despite their success. Research into resistance mechanisms uncovered a new adaptive resistance mechanism linked to JNK/c-Jun signaling. This pathway contributes to the survival of cells tolerant to alectinib and brigatinib. Blocking JNK/c-Jun improved the effectiveness of ALK-TKI treatment in curbing cell growth and promoting cell death. Combining the inhibition of JNK with ALK-TKIs increased cell death by suppressing Bcl-xL proteins, surpassing the effects observed with ALK-TKI treatment alone. Targeting both JNK signaling and ALK might be a promising method to improve outcomes for ALK-rearranged NSCLC [68]. 

Additionally, treatment advancements for NSCLC with the EML4-ALK fusion gene have been made with ALK-TKIs. In ALK-TKI-resistant cells, the expression of EML4-ALK decreased at the transcriptional level, while the phosphorylation of EGFR, HER2, and HER3 increased compared to parental-sensitive cells. This increase in the activation of HER family proteins coincided with a higher secretion of EGF. Treatment with an EGFR-TKI induced apoptosis in ALK-TKI resistant cells but not in sensitive cells. In the parental cells, the inhibition of extracellular signal-regulated kinase (ERK) and STAT3 phosphorylation by the selective ALK-TKI TAE684 was disrupted when these cells were exposed to exogenous EGF, leading to reduced sensitivity in cell growth to TAE684 [69]. However, resistance, notably the G1202R mutation in ALK, limits their effectiveness. A recent study demonstrated that the EML4-ALK G1202R mutation prompts an epithelial-mesenchymal transition (EMT), likely boosting cell migration and invasion through increased STAT3 and Slug expression. Combining ALK and STAT3 inhibitors restores sensitivity to ceritinib, offering a potential approach to counter ALK mutation-driven resistance in NSCLC therapy [70]. Nonetheless, the ALK-TKI TAE684 suppressed cell growth, triggered cell death, and blocked the activation of STAT3 and ERK in H3122 cells carrying the EML4-ALK fusion gene, but not in H2228 cells with the same fusion gene demonstrated that TAE684 predominantly inhibited STAT3 activation without significantly impacting cell growth or apoptosis. However, the combined use of TAE684 and an MEK inhibitor induced cell death by concurrently targeting the STAT3 and ERK pathways in H2228 cells. This combined inhibition reduced levels of the antiapoptotic protein survivin and increased levels of the proapoptotic protein BIM [71]. 

In addition, research exploring protein methylation, notably SET and MYND domain containing 2 (SMYD2) methyltransferases, discovered their role in methylating specific lysine residues (1451, 1455, and 1610) in the ALK protein. Lowering SMYD2 levels or using an SMYD2 inhibitor reduced EML4-ALK protein phosphorylation in NSCLC cell lines. Modification of these lysine residues hindered ALK methylation and inhibited downstream AKT phosphorylation, impeding cell growth. Combining SMYD2 and ALK inhibitors demonstrated enhanced efficacy in restraining NSCLC cell growth. Hence, this SMYD2-mediated ALK methylation is suggested as a novel treatment avenue for ALK fusion gene-related tumors [72].

A patient with EGFR mutation and EML4-ALK rearrangement, post-EGFR-TKI resistance, showed promising responses to combined EGFR and ALK inhibitors, suggesting a viable therapeutic approach for managing NSCLC with concurrent mutations. A patient with NSCLC developed both an EGFR mutation and an EML4-ALK rearrangement after resistance to EGFR-TKI treatment. Researchers engineered EGFR mutant cells with ALK variants to understand how these molecular combinations function and tested various inhibitors in laboratory and animal settings. The findings revealed that cells expressing these variants resisted individual treatments but responded positively to a combination of ALK and EGFR inhibitors, showing elevated effectiveness in killing cancer cells. In animal experiments, this combination therapy significantly reduced tumor growth compared to individual treatments. Particularly noteworthy was a patient with liver metastases experiencing a decrease in liver tumor size after receiving a combination of osimertinib (an EGFR-TKI) and ceritinib (an ALK-TKI). The study suggests that employing both EGFR and ALK inhibitors might be a promising therapeutic approach for managing NSCLC marked by simultaneous EGFR mutation and EML4-ALK rearrangement [73,74]. 

Moreover, scientists aimed to develop more potent ALK inhibitors to combat drug resistance in ALK rearrangement-related NSCLC [75,76]. They identified ZX-29 as a potent inhibitor that caused G1 phase cell cycle arrest and subsequent cell death via endoplasmic reticulum stress. Notably, ZX-29 induced protective autophagy, and blocking this process enhanced its effectiveness against tumors. Additionally, ZX-29 effectively hindered mouse tumor growth and showcased its ability to overcome drug resistance stemming from the ALK G1202R mutation [75]. Moreover, scientists developed XMU-MP-5, a new ALK inhibitor to combat crizotinib resistance in NSCLC. In laboratory and mouse model studies, XMU-MP-5 successfully targeted ALK pathways, inhibiting cell proliferation in both wild-type and mutant EML4-ALK cells. These preclinical outcomes underscore XMU-MP-5 as a promising, highly selective ALK inhibitor capable of addressing clinically relevant secondary ALK mutations [76]. An extensive analysis was also carried out on 31 cancer tissues and 90 circulating cell-free DNA (cfDNA) samples. Among cancers resistant to crizotinib, 16% displayed ALK mutations (such as L1196M, I1171T, D1203N, G1269A/F1174L) and three potential bypass mutations. Ceritinib-resistant cancers exhibited 22% ALK mutations (including G1128A, G1202R, G1269A, I1171T/E1210K) and similar bypass mutations. Alectinib-resistant cancers showed 17% ALK mutations (including G1202R, W1295C, G1202R/L1196M) and one potential bypass mutation. Lorlatinib-resistant cancers had 11% ALK mutations (including G1202R/G1269A) and two potential bypass mutations. Cases with both tissue and cfDNA samples revealed mutations in 45% and 30%, respectively, with a matching rate of 45% [77].

Eventually, researchers have discovered a direct relationship between ALK and cyclin-dependent kinase 9 (CDK9) in breast cancer, where ALK phosphorylates CDK9, leading to resistance against Poly(ADP-Ribose) Polymerase (PARP) inhibitors and encouraging homologous recombination repair. This phosphorylation boosts CDK9’s activity, promoting gene transcription linked to HR-repair within the nucleus. When ALK is inhibited, CDK9 is degraded by Skp2, an E3 ligase. These discoveries propose a treatment avenue based on specific biomarkers, combining ALK and PARP inhibitors to induce synthetic lethality in PARP inhibitor-/platinum-resistant tumors expressing elevated p-ALK-p-Tyr19-CDK9 [78]. Moreover, considering the promising suppression of NSCLC with CDK9 inhibitors seen in EGFR-TKI resistant NSCLC [79], further investigation into their potential in ALK+ NSCLC is warranted.

In conclusion, the diverse mechanisms of resistance and intricate signaling pathways associated with ALK+ cancers, particularly in NSCLC and other malignancies, reveal the complexity of targeting ALK alterations. Various studies have highlighted the signaling cascades activated by mutated forms of ALK and ALK fusion proteins, such as NPM-ALK, elucidating their role in cell transformation, survival, and resistance to therapy. Strategies involving dual inhibition of ALK and other pathways, including MYC, Src, JNK/c-Jun, and Wnt/β-catenin, have demonstrated potential in overcoming resistance and hindering tumor growth in ALK+ cancers. The discovery of ALK mutations and their association with resistance to different ALK inhibitors has spurred the development of newer, more potent inhibitors such as ZX-29 and XMU-MP-5, showcasing promising preclinical efficacy against various ALK mutations, including the challenging G1202R mutation. These findings underscore the need for a multifaceted approach to combat ALK-related resistance and advance therapeutic strategies, emphasizing the need for further investigation and clinical trials to optimize treatment outcomes in ALK+ cancers.

## 5. FDA-Approved ALK Inhibitors

### 5.1. Main Clinical Trials

Studies comparing various ALK inhibitors for advanced NSCLC have highlighted distinct efficacy profiles and safety concerns associated with each medication. Examining the efficacy of crizotinib, alectinib, brigatinib, ceritinib, and lorlatinib in treating ALK+ NSCLC has provided valuable insights into their performance, safety, and unique adverse event profiles [80].

Clinical studies [81] highlighted crizotinib’s superior performance over chemotherapy, emphasizing extended PFS and higher ORR. Similarly, another research [82] reinforced these findings in PROFILE 1029 (NCT01639001), underlining the significant advantages of crizotinib in terms of PFS, ORR, and prompt response time among East Asian patients despite negligible differences in overall survival (OS). Conversely, several studies have consistently drawn attention to alectinib’s advantages compared to crizotinib. Peters et al. [83] and Gadgeel et al. [84] independently demonstrated alectinib’s superior PFS and central nervous system (CNS) activity in untreated ALK+ NSCLC patients, irrespective of prior CNS disease. Further studies examining cfDNA as a prognostic biomarker [85] and ALK+ tumor responses [86] consistently favored alectinib over crizotinib. Additionally, patient-reported outcomes from the ALEX trial (NCT02075840) showcased alectinib’s prolonged benefits in lung cancer symptom management and superior CNS progression control [79].

The global phase III ALEX study showcased notable enhancements in PFS and OS when comparing alectinib to crizotinib in treatment-naive individuals with ALK + NSCLC. Subsequent phase III trials in Japanese and Asian populations (J-ALEX and ALESIA) affirmed the clinical advantages of alectinib over crizotinib as a first-line therapy. Alectinib demonstrated a well-managed safety profile throughout these pivotal trials, with this review concentrating on the prolonged safety and tolerability of alectinib in advanced ALK + NSCLC. Most adverse events linked to alectinib can be effectively managed through dose reduction, and the safety profile remains stable during extended follow-up, with no emergence of new signals. These findings reinforce alectinib’s position as the preferred treatment for treatment-naive advanced ALK + NSCLC [87].

On the other hand, in the ALTA-1L trial (NCT02737501), brigatinib’s effectiveness was evaluated against crizotinib among patients with locally advanced or metastatic ALK+ NSCLC who had not previously received ALK inhibitors [88]. Brigatinib demonstrated significant superiority over crizotinib in terms of PFS and ORR. The trial highlighted brigatinib’s substantial increase in response duration compared to crizotinib, with an OS rate of 86% for crizotinib and 85% for brigatinib. Notably, the adverse effect profiles differed between the two treatments, with distinct patterns of side effects observed in each group. Further analysis from the second interim assessment reinforced brigatinib’s superior PFS over crizotinib, supporting its efficacy in delaying disease progression [89]. Despite similar OS probabilities at two years, brigatinib consistently outperformed crizotinib in PFS. However, adverse effects, especially those in grades 3–5, were higher in the brigatinib group. Moreover, researchers [90] conducted a comparative analysis of the ALTA-1L trial outcomes in Asian and non-Asian patients, affirming brigatinib’s notable advantages in PFS across both subgroups, while overall safety remained comparable.

Additionally, Ng et al. [91] focused on the unique pulmonary-related adverse events associated with brigatinib among ALK inhibitors, highlighting their rarity but significance. Their assessment across phase 1 to 3 trials revealed early onset pulmonary events, albeit at a low percentage, emphasizing the importance of monitoring for these specific adverse events during brigatinib treatment. In the future, the ALTA-3 trial (NCT03596866) aims to assess how well brigatinib performs compared to alectinib in individuals with advanced ALK+ NSCLC resistant to crizotinib, offering additional information about the effectiveness and relative results of these therapies [92].

In the multicenter, randomized ASCEND-4 trial (NCT01828099) comparing ceritinib to platinum-based chemotherapy, the efficacy and safety of ceritinib in ALK-rearranged nonsquamous NSCLC were assessed [93]. Ceritinib demonstrated significantly longer PFS than chemotherapy, with substantial, rapid, and durable responses observed in the ceritinib group. However, adverse events, particularly diarrhea, nausea, and vomiting, were more familiar with ceritinib, including higher-grade events such as elevated liver enzymes. The effectiveness and tolerability of ceritinib were further analyzed in a Japanese subgroup of patients from the ASCEND-5 (NCT01828112) study [21,94]. The ceritinib group exhibited prolonged PFS compared to chemotherapy, though it came with a higher incidence of suspected drug-related adverse events. Another comparison, presented by Li et al. [81], evaluated the outcomes of PROFILE 1014 (NCT01154140) and ASCEND-4 (NCT01828099) phase 3 trials. Ceritinib significantly improved PFS compared to crizotinib, showcasing a notable reduction in disease progression or mortality risk. Despite a comparable OS rate at 12 months, ceritinib exhibited a clinically meaningful advantage by maintaining a higher PFS rate. This analysis indicated a substantial improvement in the treatment of first-line metastatic NSCLC. In the ALUR trial (NCT02604342), a randomized, multicenter, open-label, phase 3 study, researchers compared the efficacy of alectinib to chemotherapy in patients with advanced or metastatic ALK+ NSCLC. This study specifically enrolled 107 patients from Europe and Asia who had previously undergone platinum-based doublet chemotherapy and crizotinib. Novello et al. [24] summarized the findings of the ALUR trial, revealing substantial improvements in PFS with alectinib compared to chemotherapy. Patients treated with alectinib experienced significantly longer PFS durations than those receiving chemotherapy (9.6–7.1 months versus 1.4–1.6 months, *p* < 0.001). Additionally, adverse events of grade 3 or higher and those leading to discontinuation of the study drug were less prevalent in the alectinib group. Furthermore, the duration of alectinib treatment was notably prolonged compared to chemotherapy (20.1 weeks versus 6.0 weeks). Moreover, a comparative analysis revealed a notable advantage of alectinib, demonstrating a prolonged PFS (68.4 months) in contrast to crizotinib (48.7 months), signifying its superior efficacy (hazard ratio range 0.15–0.47). Alectinib consistently outperformed crizotinib across all secondary outcomes, including ORR [95].

In the CROWN clinical trial (NCT03052608), a phase 3, open-label, multicenter, randomized study, researchers investigated the efficacy of lorlatinib as a first-line treatment for advanced ALK+ NSCLC in comparison to crizotinib [96]. The trial enrolled 296 patients without previously received systemic therapy for their metastatic disease. Patients were randomized to receive either lorlatinib (100 mg daily) or crizotinib (250 mg twice daily) for 28 days. The study revealed significant advantages in favor of lorlatinib over crizotinib. The proportion of patients alive at 12 months without disease progression was notably higher in the lorlatinib arm compared to the crizotinib arm (78% versus 39%, *p* < 0.001). Additionally, lorlatinib demonstrated superior PFS within one year compared to crizotinib (80% versus 35%). Objective responses were observed in 76% of patients in the lorlatinib group and 58% in the crizotinib group. The most common adverse effects associated with lorlatinib were hyperlipidemia and edema, while crizotinib was linked to different adverse effects. Assessing the quality of life (QoL), both groups experienced overall improvements and delayed declines [97]. Crizotinib showed better results in cognitive functioning, while lorlatinib demonstrated advantages in physical, role, emotional, and social functioning. The lorlatinib group showed more noticeable enhancements in symptoms such as fatigue, nausea, vomiting, insomnia, appetite loss, constipation, diarrhea, and congestion, whereas crizotinib was more beneficial for improving peripheral neuropathy [98].

### 5.2. Efficacy and Tolerability Profiles

In a clinical study involving 72 Chinese patients diagnosed with ALK+ NSCLC, crizotinib demonstrated favorable effectiveness and was well-tolerated. Administered orally at 250 mg twice daily, the patients, primarily characterized as young, non/light smokers with adenocarcinoma histology, achieved an ORR of 52.2% and a disease control rate of 64.2%. Common adverse effects included mild visual disturbances, nausea, vomiting, diarrhea, and constipation. The findings suggest that crizotinib is well-tolerated and effective in this patient cohort, highlighting the need for further prospective, multicenter studies with larger sample sizes to validate these results [99].

Moving to the NP28761 phase II study conducted in North America (NCT01871805), alectinib demonstrated notable efficacy and well-tolerated outcomes in individuals diagnosed with ALK + NSCLC. Alectinib exhibited swift efficacy, with a median time to symptom improvement of 1.4 months and a median time to symptom deterioration of 5.1 months. Patients with baseline CNS metastases maintained QoL comparable to those without CNS metastases throughout the study. Patients treated with alectinib in this study witnessed substantial enhancements in QoL and symptom relief, coupled with a delayed onset of symptom deterioration [100]. Transitioning to another study involving 207 patients with ALK + NSCLC, a comparison was made between alectinib and crizotinib. The second interim analysis unveiled superior efficacy with alectinib, showcasing a median PFS not reached, while crizotinib exhibited a median PFS of 10.2 months. However, discontinuation rates were lower in the alectinib group (24 patients) compared to the crizotinib group (61 patients), primarily due to lack of efficacy or adverse events. Grade 3 or 4 adverse events and dose interruptions due to adverse events were more prevalent with crizotinib, and a higher number of patients in the crizotinib group discontinued the study drug due to adverse events. No adverse events with a fatal outcome occurred in either group. These findings, constituting the inaugural head-to-head comparison of alectinib and crizotinib, can impact the standard of care for first-line treatment in ALK + NSCLC [101].

Furthermore, lorlatinib, identified as a potent and highly active ALK inhibitor with favorable outcomes in later-line settings, aligns with clinical trial data. In a real-world evaluation of 38 heavily pretreated patients with ALK + NSCLC, lorlatinib demonstrated notable efficacy and tolerability. The overall response rate was 44%, and the disease control rate was 81%, indicating substantial antitumor activity. Lorlatinib dose adjustments, including reduction (18%), interruption (16%), and discontinuation (3%), were consistent with the trial experience. Median OS from advanced ALK+ diagnosis ranged from 45.0 to 69.9 months, while median PFS from lorlatinib initiation varied from 7.3 to 27.7 months. Notably, patients with brain metastases showed a trend toward improved median PFS compared to those without (34.6 months vs. 5.8 months). Lorlatinib exhibited a median intracranial PFS of 14.2 months [102]. Additionally, the effectiveness and well-tolerated nature of entrectinib in addressing gene fusions related to tyrosine kinases TRKA/B/C, ROS1, and ALK in solid tumors, including those affecting the CNS, were documented. Entrectinib exhibited outstanding tolerability, primarily with reversible grade 1/2 adverse events. Substantial antitumor responses were noted in diverse cancers, including NSCLC. Particularly noteworthy was a comprehensive CNS response in a patient with sequestosome 1-neurotrophic receptor tyrosine kinase (1SQSTM1-NTRK1)-rearranged lung cancer. Most treatment-related adverse events were grade 1/2 and manageable/reversible with dose modifications. Discontinuation rates related to treatment-related adverse events occurred in 8.3% of patients [103,104].

In conclusion, examining various ALK inhibitors for advanced NSCLC has unveiled their distinct efficacy profiles and safety considerations. While crizotinib showcased notable advantages regarding PFS and ORRs, alectinib consistently emerged as a superior performer across multiple studies. Alectinib’s prolonged PFS heightened CNS activity, and excellent patient-reported outcomes stood out prominently. Despite its effectiveness in extending PFS, brigatinib exhibited more adverse effects than crizotinib. Ceritinib’s ability to improve PFS and reduce the risk of disease progression or mortality underscored its clinical significance. As a first-line treatment, lorlatinib demonstrated promising outcomes, displaying superior PFS and a higher proportion of patients without disease progression at the 12-month than crizotinib. However, it is essential to note that lorlatinib showed varying adverse effect profiles and impacts on different facets of patients’ quality of life. These findings provide clinicians with invaluable insights to tailor therapies according to individual patient needs and tolerability, ultimately advancing the development of personalized treatment strategies for ALK+ NSCLC.

## 6. Advancements in ALK-Targeted Therapy and Future Horizons

### 6.1. ALK and Immunotherapy

Changes in the structure of the *ALK* gene significantly contribute to the onset of different human cancers, and therapies aimed at this gene have revolutionized how we treat these tumors driven by this specific oncogene. However, overcoming inherent or acquired resistance remains a significant hurdle. Variations in the *ALK* gene, such as gene rearrangements or mutations, also influence the immune environment within tumors. Harnessing immunotherapy to target the *ALK* gene holds promise in clinical settings [10]. The association between ALK rearrangement and immune cells is complex and contingent on the specific characteristics of the tumor microenvironment. Generally, the presence of ALK rearrangement in cancer cells has the potential to influence the dynamics of interactions between tumor and immune cells. Some studies propose that ALK+ cancers may exhibit distinct immunological characteristics compared to their ALK-negative counterparts. Moreover, the tumor microenvironment plays a pivotal role in shaping the immune response against cancer.

In ALK + NSCLC, the proportion of tumors expressing PD-L1 was lower compared to KRAS + NSCLC. Furthermore, T cells expressing immune checkpoint proteins, including T-cell immunoglobulin and mucin-domain containing-3 (TIM-3), CTLA4, Lymphocyte activation gene 3 (LAG3), and PD-1, were less prevalent in ALK+ NSCLC than in EGFR/KRAS + NSCLC. Additionally, the levels of CD3, CD8 T cells, and CD20 B cells were lower in ALK+ NSCLC compared to KRAS+ NSCLC, while CD4 helper T cell levels were higher in ALK+ NSCLC than in EGFR/KRAS+ NSCLC. TIM3 repression was higher in ALK+ NSCLC than in KRAS+ NSCLC. Notably, high expression of PD-L1 and CTLA4 was associated with lower OS in advanced-stage ALK-rearranged NSCLC patients treated with ALK-TKIs. These findings suggest an immunosuppressive tumor microenvironment in ALK + NSCLC, emphasizing the need for further exploration and validation of immunotherapy in this patient population through clinical trials [105]. Additionally, individuals with EGFR mutations or ALK rearrangements exhibited the lowest proportion of tumors expressing both PD-L1 and CD8 (PD-L1+/CD8+), at 5.0%. In contrast, at 63.5%, the highest proportion was observed in tumors lacking both PD-L1 and CD8 expression (PD-L1-/CD8-). Conversely, those with wild-type EGFR and ALK presented 14.2% of tumors showing PD-L1+/CD8+ and 50.3% with PD-L1-/CD8-. Consequently, patients harboring EGFR mutations or ALK rearrangements demonstrated a diminished PD-L1 and CD8 co-expression level in the tumor microenvironment, potentially contributing to an inadequate response to ICIs. The co-expression of PD-L1 and CD8 in EGFR-mutated or ALK-rearranged lung cancer serves as a biomarker for poor prognosis, correlating with a shorter OS [106]. Moreover, in an investigation involving 12 patients with stage IIA-IIIB NSCLC undergoing neoadjuvant ALK-TKI therapy, a notable ORR of 91.7% and a major pathological response rate of 75.0% were documented. Specifically, 58.3% of patients achieved a pathological complete response. After neoadjuvant ALK-TKI administration, a substantial increase in the immune infiltration of CD8+ and CD4+ cells was observed. Conversely, macrophages, encompassing M1 and M2 subtypes, exhibited minimal changes post-therapy. These findings suggested that neoadjuvant ALK-TKI treatment is safe and viable for ALK+ resectable NSCLC, yielding favorable pathological responses and influencing the tumor immune microenvironment [107].

On the other hand, interleukin (IL)-6, IL-8, and IL-10 have been associated with disease progression in NSCLC, specifically in ALK+ patients. The interactions between TLRs and various interleukins underscore their involvement in lung cancer pathogenesis, progression, and potential prognostic value [108]. Moreover, serum-soluble IL-2R levels are a reliable marker for disease activity in hairy cell leukemia and adult T-cell leukemia/lymphoma patients. In addition, ALCL patients often display CD30 and CD25 expression in malignant cells. A study measured serum soluble IL-2R and CD30 levels in ALCL patients treated with etoposide, prednisone, Oncovin, Cytoxan, and hydroxydaunorubicin (EPOCH) chemotherapy. Soluble CD30 levels were initially high and decreased with treatment. This study also demonstrated that patients with the ALK gene had higher soluble IL-2R levels than those without ALK, whose soluble IL-2R levels were normal and whose tumors lacked CD25 expression. Elevated soluble IL-2R levels were observed in patients with recurrent disease, regardless of ALK status [109].

Further, the research explored lymphoma immune profiles, which are crucial for accurate diagnosis and new treatment avenues. In ALCL, microRNA-135b (miR-135b), influenced by the NPM-ALK oncogene, promoted cancer growth and instigated an immune profile producing IL-17. NPM-ALK activated miR-135b via STAT3, targeting FOXO1 and impacting ALCL cells’ chemotherapy response. Additionally, miR-135b hindered Th2 regulators STAT6 and GATA3, altering IL-17 production and resembling the immune profile of Th17 cells in ALCL. Blocking miR-135b reduced tumor growth and blood vessel formation in experiments, indicating its potential as a therapeutic target. This highlights how cancer-causing pathways affect tumor immune profiles and the surrounding environment [110]. 

Furthermore, the PD-1/PD-L1 axis in ALK+ tumors favors regulatory T cells (Tregs), limiting T cell activity. Additionally, PD-L1 upregulation promotes an immunosuppressive environment, hindering effective antitumor responses. Meanwhile, ALK release into the tumor microenvironment stimulates tumor-associated macrophages (TAM) and B-cells, shaping a milieu conducive to tumor growth and immune evasion. Understanding these dynamics is pivotal in developing targeted therapies against ALK-driven tumors, aiming to recalibrate the immune landscape and disrupt tumor-promoting interactions. Alternatively, in ALCL patients, specific levels of serum cytokines could indicate the tumor’s size and variations in the body’s response against the lymphoma. Levels of IL-9, IL-10, IL-17a, HGF, soluble IL-2R, and soluble CD30 collectively create a distinct cytokine profile specific to ALK+ ALCL, distinguishing them from both remission samples and samples from children of similar age with low-stage B-cell non-Hodgkin lymphoma, serving as special control groups. Furthermore, cytokine levels such as IL-6 and interferon (IFN)-γ correlated with the disease stage, patient condition, and the likelihood of relapse among ALCL patients, with IL-6 displaying individual predictive value. These initial cytokine profiles in ALCL patients might reflect the tumor characteristics and the strength of their immune responses [111]. Furthermore, the researchers explored how circulating cytokines might be markers for monitoring disease progression in ALK+ NSCLC when treated with TKIs. They examined eight cytokines in serum samples from 38 patients. Higher levels of IL-6, IL-8, and IL-10 correlated with disease advancement, particularly IL-8, which displayed the most substantial potential as a biomarker. Although the combination of alterations in IL-8 alongside circulating tumor DNA parameters enhanced the ability to detect disease progression, it did not exceed the accuracy achieved by using circulating tumor DNA alone. This study suggested that serum cytokine levels might indicate disease progression in ALK+ NSCLC, potentially enhancing current monitoring methods pending further confirmation in more extensive studies (Figure 6) [112].

The occurrence of any mutation in ALK results in the promotion of PD-L1 expression. Increasing the expression of immunosuppressive molecules such as PD-L1 may lead to tolerance and immune evasion in patients with tumors and cancers. Tian et al. have shown that upregulation of PD-L1 can be identified as a biomarker for ALK-rearrange NSCLC. In addition, it has been recognized that the TME in the presence of upregulated expression of PD-L1 encompasses an immunosuppressive condition [113,114]. ICIs have shown significant promise in various cancers [115,116]. In ALK+ NSCLC, these inhibitors have been explored, mainly due to the upregulation of PD-L1 expression in ALK+ tumors [117,118]. However, studies on the prognosis of ALK+ patients using ICIs have yielded conflicting results, necessitating further investigation [117]. Initial data from randomized studies suggested lower effectiveness of immunotherapies in ALK+ tumors compared to wild-type tumors. For instance, a global retrospective study found no objective response in ALK+ NSCLC patients treated with ICI monotherapy, with a higher incidence of rapid progression in this group. Subsequent investigations in this area mainly focused on understanding ALK inhibitor resistance [119].

In addition, another study studied how ALK fusion proteins regulate PD-L1 expression and immune function in ALK+ NSCLC. Researchers observed a correlation between PD-L1 expression, EGFR mutations, and ALK fusion genes in NSCLC cell lines. Elevating ALK fusion protein levels boosted PD-L1 expression, leading to T-cell apoptosis in co-culture systems. Blocking ALK with TKIs amplified IFN-γ production. Anti-PD-1 antibodies were effective in both crizotinib-sensitive and -resistant NSCLC cells. However, combining ALK-TKIs with anti-PD-1 antibodies did not benefit co-culture systems. ALK-TKIs suppressed tumor growth and indirectly bolstered antitumor immunity by reducing PD-L1 expression. While anti-PD-1/PD-L1 antibodies could be an option for ALK+ NSCLC patients, especially crizotinib-resistant ones, combining ALK-TKIs with anti-PD-1/PD-L1 antibodies requires further study before clinical use [114].

On the other hand, attempts to combine ICIs with ALK-TKIs showed some promise. Preclinical research demonstrated that combining ceritinib (an ALK inhibitor) with a PD-L1 inhibitor suppressed PD-L1 expression and enhanced lymphocyte activity in ALK-rearranged NSCLC [120]. However, several phase 1/2 clinical trials combining nivolumab or pembrolizumab (PD-1 inhibitors) with crizotinib or lorlatinib (ALK inhibitors) reported dose-limiting toxicities, impacting their efficacy. A few studies indicated potential benefits with ceritinib plus nivolumab, suggesting activity, especially in patients with high PD-L1 expression [120,121,122]. Furthermore, combining avelumab or atezolizumab (PD-L1 inhibitors) with lorlatinib or alectinib (ALK inhibitors) demonstrated promising efficacy in ALK+ NSCLC. The use of alectinib combined with atezolizumab showed promising results, but heightened toxicity was observed compared to using each agent separately. Due to limited sample sizes and relatively short observation periods, definitive conclusions about the treatment’s effectiveness against tumors could not be drawn. Nonetheless, more extensive studies are required to verify these outcomes further [123]. However, the level of PD-L1 expression may not reliably indicate the expected response to initial treatment with alectinib in patients with ALK+ NSCLC [124].

Furthermore, the combination of ALK-targeted therapy and ICIs for ALK-modified NSCLC is being clinically investigated. Preclinical data initially supported the nivolumab (PD-1 inhibitor) and crizotinib combination, demonstrating increased PD-L1 expression in the presence of ALK-EML4 fusion protein and how both checkpoint and ALK inhibitors reduced T cell apoptosis and crizotinib-resistant cell survival [114]. Despite this promise, adverse events in the trial obstructed the evaluation of the combo’s efficacy, underscoring the necessity for robust phase 1 studies despite encouraging preclinical evidence. In a phase 1/2 study (NCT02574078), the combination of nivolumab and crizotinib was examined as a first-line treatment for advanced NSCLC with ALK translocation. This study aimed to assess the safety and tolerability of nivolumab in various NSCLC treatment settings. Patients with confirmed ALK-translocation positive NSCLC were administered nivolumab intravenously every two weeks alongside crizotinib orally twice daily. However, during the interim safety review, 23% of the initial 13 treated patients encountered grade 3 or more hepatic toxicities, leading to the discontinuation of the combination treatment. One patient suffered from grade 4 treatment-related pneumonitis, grade 3 rhabdomyolysis, and grade 3 alanine aminotransferase (ALT)/aspartate aminotransferase (AST) elevations, with pneumonitis cited as the cause of death. The second patient had treatment-related grade 4 acute liver failure, contributing to acute respiratory failure and disease progression, culminating in death. Although the combination showed partial responses in 38% of patients, the trial did not meet its primary safety endpoint, prompting the authors to advise against further exploration of this combination for ALK-translocation NSCLC [125,126]. However, discontinuation rates due to hepatic adverse events were generally low with single-agent nivolumab and crizotinib in the treatment of advanced NSCLC (NCT02393625, NCT01642004, NCT01673867) [119,127].

Nevertheless, the study (CheckMate-370), where nivolumab and crizotinib were combined for the primary treatment of ALK translocation–positive advanced NSCLC, did not meet the anticipated number of patients with an objective response. Despite regular monitoring of liver function tests every two weeks during the study, researchers observed a higher rate of treatment discontinuation due to hepatic treatment-related adverse events with the combination of nivolumab and crizotinib regimen than historical expectations based on observations with monotherapy alone. This outcome was below the anticipated level based on responses to crizotinib monotherapy. A notable rate of early discontinuation likely undermined the efficacy outcomes. The specific mechanisms underlying the observed toxicities in this combination are currently unknown. Possible explanations include additive effects, drug-drug interactions leading to the creation of reactive drug metabolites that interfere with cell function and induce cell death, off-target effects, exacerbation of tyrosine kinase inhibitor-induced damage by checkpoint inhibitors, or immune-based effects, among other possibilities [126].

Another path being explored is the combination of ICIs with anti-angiogenesis therapy. While atezolizumab combined with chemotherapy did not exhibit enhanced survival compared to chemotherapy alone in ALK+ patients, the combination of atezolizumab with bevacizumab, an anti-vascular endothelial growth factor (VEGF) agent, along with chemotherapy demonstrated marked improvements in PFS and hinted at a potential benefit for OS [128,129]. This combined approach might be favored for ALK+ NSCLC patients’ post-resistance to ALK inhibitors. Studies scrutinizing the synergy between ICIs and anti-VEGF agents have underscored their combined mechanisms, indicating the potential to counteract resistance to ICIs by reversing VEGF-mediated immunosuppression and bolstering the response to ICIs in ALK+ tumors. Overall, the confluence of ICIs with anti-angiogenesis therapy presents a promising treatment strategy for ALK+ NSCLC [130].

Angiogenic factors create vascular abnormalities and hinder antigen presentation, suppress immune cells, and boost cell activity that inhibits the immune system. ALK signaling in the PI3K/AKT/mTOR pathway stimulates VEGF expression, potentially heightening sensitivity to bevacizumab in ALK+ patients. Following treatment with ALK inhibitors, individuals with ALK+ tumors encounter a decline in the infiltration of CD8+ T cells and a rise in regulatory T cells, resulting in a diminished response to ICIs [131]. Moreover, clinical studies indicate that combining bevacizumab and atezolizumab overcomes ICI resistance by reversing VEGF-induced immunosuppression and enhancing CD8+ tumor-infiltrating lymphocytes (TIL) in tumors. Evidence suggests that bevacizumab can overcome ALK inhibitor resistance when combined with targeted therapy. Recent research highlights VEGFR2 inhibition as a promising strategy for inhibiting tumor angiogenesis and directly impeding cancer cell growth in oncogene-driven NSCLC [132]. However, two patients with ALK-rearranged lung cancer, previously treated with multiple ALK-TKIs and other therapies, showed promising outcomes when treated with a combination of bevacizumab and lorlatinib. They experienced disease regression and control for 5–9 months, surpassing the duration of single-agent erlotinib therapy. This combination was well tolerated and could benefit patients after lorlatinib failure, especially against both on-target (such as ALK kinase domain mutations) and off-target resistance mechanisms. Additionally, it might hold promise for patients before lorlatinib treatment or those with ROS1-rearranged lung cancers. The study suggests exploring this combination further in clinical settings for ALK+ lung cancer patients [133,134].

### 6.2. Molecular Diagnosis of ALK: Insights from Next-Generation Sequencing

Molecular analyses, mainly focusing on genetic rearrangements in genes such as ALK, ROS1, RET, and NTRK [135], have become standard practices in patients with advanced NSCLC—immunohistochemistry (IHC) functions as the primary screening method, valued for its ease of implementation and interpretation. Fluorescence in situ hybridization (FISH) confirms rearrangements, especially in cases with ambiguous immunostainings. Although FISH is acknowledged as the most sensitive method for detecting ALK and ROS1 rearrangements, it requires comprehensive guidelines for result interpretation [136].

On the other hand, advanced genomic analyses, such as next-generation sequencing (NGS), meticulously scrutinize the genetic composition of NSCLC. The pivotal roles of ALK, ROS1, and RET genes in NSCLC development make their fusion events crucial for targeted therapies. Researchers characterize these fusion events by employing cutting-edge techniques, aiming for a profound understanding of the molecular intricacies driving NSCLC progression. The expanding coverage of genetic testing has led to the discovery of numerous ALK fusion subtypes and partners, with over 90 rare ALK fusion subtypes identified in NSCLC. While common fusions such as EML4-ALK have established clinical data, rare fusions such as striatin (STRN)-ALK and huntingtin interacting protein 1 (HIP1)-ALK lack substantial clinical evidence. ALK-TKIs are clinically applied based on ALK gene positivity, irrespective of the fusion partner [137,138].

The research utilized target-capture DNA NGS to identify ALK, ROS1, and RET fusions in NSCLC, examining genomic breakpoints as predictors of targeted therapy efficacy. Categorizing canonical and uncommon fusions among 3787 samples based on breakpoint positions, RNA sequencing revealed 12.8% of uncommon fusions as nonproductive. The study stressed unreliable efficacy prediction for uncommon genomic breakpoints, recommending RNA or protein validation [139]. In another investigation, a hybridization-based NGS approach on 302 NSCLC tumors identified three non-EML4-ALK fusions and additional fusions through RNA sequencing, emphasizing NGS as promising for ambiguous cases and novel fusion detection [140]. Moreover, NGS analysis of 55 patients with ALK fusion identified 92% with recognized EML4-ALK fusion variants. Sequential ALK inhibitor administration showed consistent outcomes, with the V1 (E13A20) variant correlating with improved PFS on crizotinib, providing insights into rare chromosomal events influencing outcomes [141].

In a study using DNA/RNA-based NGS and RT-PCR on tissues from 153 individuals, researchers found a high concordance rate of 95.16% between NGS and RT-PCR for identifying EML4-ALK fusion in NSCLC patients, suggesting that RNA-based approaches may offer better precision, benefiting clinical applications in NSCLC diagnosis and treatment [142]. However, NGS analysis of 155 ALK/RET/ROS1-rearranged NSCLC patients revealed TP53 mutations and cyclin-dependent kinase inhibitor 2A/B (CDKN2A/B) copy number loss, impacting the tumor immune microenvironment and clinical outcomes. Notably, the patients with TP53 or CDKN2A/B co-occurrence exhibited an immunosuppressive microenvironment, with higher PD-L1 expression but lower levels of CD8+ TILs, and experienced a worse prognosis. The findings highlight the complex interplay between genomic alterations, the tumor immune microenvironment, and clinical outcomes in ALK/RET/ROS1-rearranged NSCLC using NGS [143].

Nevertheless, a study evaluated whether variant allele frequencies (VAFs) of ALK fusions, assessed by NGS, predicted intratumoral heterogeneity (ITH) and targeted therapy efficacy in NSCLC. Among 4548 patients, 7.2% were ALK+, with no significant correlation between ALK subclonality and crizotinib efficacy, suggesting unreliable NGS-based prediction in NSCLC [144]. In another study comparing ALK detection methods, NGS demonstrated higher sensitivity than FISH, emphasizing IHC’s role in diagnosing ALK rearrangements despite NGS advantages [145]. Interestingly, a phase 2 study in China using NGS on blood samples demonstrated lorlatinib’s efficacy against ALK compound mutations, suggesting strategies to overcome resistance [146]. Similarly, in the ASCEND-1 study, NGS on tumor biopsies elucidated ceritinib’s response and resistance mechanisms, highlighting NGS’s potential in guiding treatment decisions [147]. Furthermore, in the Blood First Assay Screening Trial’s ALK+ cohort, blood-based NGS showcased its clinical utility in guiding treatment decisions for ALK+ NSCLC [148].

On another note, cfDNA-NGS analysis helps identify resistance mechanisms to ALK-targeted therapy in ALK+ NSCLC. A study enrolled 92 ALK+ NSCLC patients, utilizing plasma cfDNA NGS for longitudinal monitoring, correlating the absence of baseline circulating tumor DNA (ctDNA) with longer PFS and OS [149]. Liquid biopsies from 24 advanced ALK+ NSCLC patients progressing on ALK inhibitors, analyzed by NGS and digital PCR, identified ALK locus resistance mutations in 38.5% of plasma samples, underscoring the potential of liquid biopsy NGS in uncovering diverse resistance mechanisms and aiding therapy decisions for ALK+ NSCLC patients [150].

Based on 3000 patients, the pivotal role of NGS-based genetic profiling in diagnosing advanced NSCLC was previously highlighted. Automated extraction of DNA and RNA from tissue samples, followed by parallel sequencing, reveals that 27% of patients are eligible for approved therapies targeting EGFR, BRAF, ALK, and ROS1. An additional 7% could benefit from experimental compounds targeting MET, ERBB2 (HER2), and RET alterations. Co-mutations and precise identification of fusion partners in translocations such as ALK and ROS1 provide valuable prognostic insights. The diagnostic approach is reliable, with low dropout rates and fast turnaround times, showcasing its practicality in personalized care and research [151]. Three recurring acquired mutations in ALK—specifically I1171T/N/S, V1180L, and G1202R—have been identified as contributors to resistance by reducing drug binding. In detail, I1171T/N/S induces structural alterations in the C-helix, resulting in diminished drug interaction, while V1180L and G1202R lead to resistance through steric hindrance. Despite these hurdles, research indicates that alternative ALK-TKIs, such as ceritinib, brigatinib, and lorlatinib, may exhibit efficacy against these mutations specific to alectinib resistance. Additionally, MET amplification is recognized as an off-target resistance mechanism in ALK+ NSCLC, with crizotinib demonstrating potential activity owing to its dual anti-ALK and anti-MET effects. The resistance landscape in ALK+ NSCLC is multifaceted, encompassing alterations such as YES1 amplification [152].

Emphasizing the need for repetitive molecular profiling, particularly in liquid biopsies and NGS analysis, is crucial for validating targets and optimizing second-line therapies following disease progression. However, considering the link between TP53 mutations in NSCLC and diminished responsiveness to EGFR, ALK, and ROS1 targeted therapy in diverse studies, the discourse delved into the influence of TP53 mutations on treatment resistance. While assessing TP53 mutation status seems logical for patient stratification in clinical decision-making, the precise clinical significance of TP53 co-mutations in ALK+ NSCLC remains unclear [153].

Furthermore, the Lung Cancer Early Molecular Assessment (LEMA) trial evaluated diagnostic scenarios for metastasized NSCLC. Comparing tissue biopsy alone (scenario 1) to scenarios involving cfDNA either first or after biopsy (scenarios 2 and 3), scenarios 2 and 3 demonstrated faster molecular profiling, identifying more targets at marginal extra costs and reducing tissue biopsies. Scenario 1 had 84% clinically relevant results, while scenarios 2 and 3 reached 93%. Mean costs were €2304 in Scenario 1, €3218 for Scenario 2, and €2448 for Scenario 3; nonetheless, Scenarios 2 and 3 decreased tissue biopsies by 16% and 9%, respectively [154]. Conversely, efforts were made to improve predictive capabilities for circulating tumor cells (CTC)in advanced NSCLC. Combining the CellSearch assay with an expanded cytokeratins profile, the assay was tailored to detect a broader range of cytokeratins. In a prospective, multicenter study, the expanded profile identified a notably higher number of CTC+ patients than the standard assay. Integration of the expanded profile enabled the quantification of EML4-ALK fusion protein expression in CTC, demonstrating correlations with PFS and OS. These integrated assays promise to enhance NSCLC patients’ treatment decisions [155].

Incorporating advanced genomic analyses and utilizing various NGS applications offers a detailed understanding of NSCLC’s complex genetic makeup. While the associated costs may be negotiable with patients, these highlighted studies collectively advance precision medicine. They provide opportunities to refine therapeutic strategies, ultimately improving outcomes for ALK, ROS1, and RET fusion-driven NSCLC patients (Figure 7).

### 6.3. ALK and Co-Targeting Approaches

The continuous evolution of ALK inhibitors has significantly improved PFS in NSCLC. Notably, second- and third-generation inhibitors such as brigatinib and lorlatinib exhibit remarkable efficacy in controlling brain metastases. The shift toward personalized medicine, involving genetic panels for diagnosis and tailored targeted therapies, represents a new paradigm. Adopting broad molecular panels as the standard of care will facilitate the detection of resistance mechanisms. This prolonged PFS is anticipated to transform the disease into a manageable, chronic condition. Effective treatment sequencing will be vital for patient survival, and the potential replacement of tissue biopsies with liquid biopsies is on the horizon [156]. In addition, clinical trials have shown that ALK inhibitors exhibit excellent efficacy against brain metastases. Consequently, initiating treatment with these specific inhibitors is considered reasonable in asymptomatic patients. Radiotherapy can then be employed during tumor progression or when symptoms arise, ensuring the best possible quality of life for patients [157,158]. Nonetheless, despite these improvements, the emergence of cancers with compound resistance mutations poses a challenge, indicating the necessity for developing multiple ALK inhibitors targeting various compound mutations. Another promising avenue is the exploration of drug combinations, where an ALK inhibitor is paired with a drug targeting a “second driver” to overcome resistance (Figure 7) [159].

Lineage transformation, recognized as a resistance mechanism to ALK-TKIs, occurs at a low frequency, less than 5%, and is primarily attributed to changes in transcriptional patterns rather than acquiring new genomic mutations in the cells [160]. In cases of resistance to second-generation ALK-TKIs, treatment strategies should be personalized according to the identified resistance mechanisms. Lorlatinib is the preferred option for patients with ALK mutations resistant to these TKIs, providing comprehensive coverage, including mutations such as G1202R and L1196M. For situations without specific resistance mutations, alternative options such as atezolizumab, bevacizumab, and platinum-based chemotherapy may be explored [161]. In cases of oligo-progression, the approach may involve maintaining the existing systemic treatment despite progression, accompanied by adding local therapies to address advancing lesions. Strategies to counteract on-target resistance mechanisms in ALK-TKI resistance include developing 4th generation TKIs (such as TPX-0131 and NVL-655) and proteolysis-targeting chimeras (PROTACs). In off-target (ALK-independent) resistance cases, potential options include combination therapies targeting ALK along with other downstream or parallel pathways, novel antibody-drug conjugates, or combining ALK inhibitors with chemotherapy and immunotherapy [162].

Crizotinib exhibits notable efficacy in ALK+ lung cancers, but variable responses and acquired resistance pose challenges. Clinical observations of an exceptional response to an insulin-like growth factor 1 receptor (IGF-1R)-specific antibody in an ALK+ patient led to the identification of therapeutic synergism between ALK and IGF-1R inhibitors. ALK fusion proteins bind to insulin receptor substrate 1 (IRS-1), and inhibiting IRS-1 enhances ALK inhibitors’ antitumor effects. In models of ALK-TKI resistance, activation of the IGF-1R pathway is observed, and combined ALK and IGF-1R inhibition improves therapeutic efficacy. Biopsy samples from patients progressing on crizotinib monotherapy show increased levels of IGF-1R and IRS-1, suggesting a role for the IGF-1R-IRS-1 pathway in both ALK-TKI-sensitive and ALK-TKI-resistant states, supporting further clinical development of dual ALK and IGF-1R inhibitors [163].

An extraordinary responder in a trial employing erlotinib and IGF-1R antibody unveiled a synergistic impact between ALK and IGF-1R inhibitors. Despite the initial unresponsiveness of the patient’s tumor to erlotinib alone, a remarkable 17-month response emerged with the combination. As subsequent molecular profiling identified an ALK rearrangement, the study proposed the IGF-1R–IRS-1 signaling axis as a potential therapeutic focus in ALK+ lung cancer, providing insights for upcoming clinical trials [164]. Nonetheless, a recent study demonstrated that many neuroblastoma (NB) cell lines exhibit IGF-1R activity, and inhibiting IGF-1R leads to decreased cell proliferation in ALK-driven NB cells. Additionally, combined inhibition of ALK and IGF-1R produces synergistic anti-proliferation effects, particularly in ALK-mutated NB cells. Mechanistically, ALK and IGF-1R significantly contribute to activating downstream PI3K-AKT and RAS-MAPK signaling pathways in ALK-mutated NB cells. The study suggested a potentially crucial role of IGF-1R in ALK-mutated NB, proposing that co-targeting ALK and IGF-1R may be advantageous in the clinical treatment of ALK-mutated NB patients [165].

Furthermore, co-targeting primary anticancer targets and corresponding drug escape pathways might enhance anticancer therapeutics. The clinical status and targets of 23 approved and 136 clinical trial multi-target anticancer drugs, focusing on co-targeting ALK, EGFR, HER2, Abl, VEGFR2, mTOR, PI3K, MEK, KIT, and DNA topoisomerase, demonstrated that the majority of approved (73.9%) and phase 3 (75.0%) drugs, as well as a significant portion of phase 2 (62.8%) and phase 1 (53.6%) drugs, co-targeted cancer drug escape pathways, suggesting a potential clinical advantage in co-targeting anticancer targets and drug escape pathways, encouraging further exploration of this strategy [166]. However, examining the molecular underpinnings of ALK oncogene dependence, scientists pinpointed the RAS-RAF-MEK-ERK pathway as essential for the survival of ALK+ tumor cells. Early co-targeting of ALK and MEK showcased improved responses and delayed resistance in preclinical ALK+ tumor models, proposing a hopeful approach to enhance the treatment of ALK+ patients [167].

On the other hand, a study explored the potential of co-targeting ALK+ lung cancer cells with alectinib and the SHP2 inhibitor SHP099. The results indicated that the combination significantly reduces cell viability in ALK+ lung cancer cells, leading to G1 cell cycle arrest and increased apoptosis. This synergistic effect is attributed to the suppression of downstream RAS/MAPK signaling and the modulation of key mediators involved in the intrinsic apoptotic pathway and cell cycle regulation, including Bim, cleaved caspase-3, cyclin D1, cyclin B1, and phosphorylated CDK1 [168]. However, the RAS-RAF-MEK-ERK pathway, targeted by MEK inhibitors such as trametinib, is crucial in cancer progression. While effective in RAS-mutated NB, ALK-addicted NB cells showed increased AKT and ERK5 activation upon trametinib. This feedback response discouraged MEK inhibitors for ALK-addicted NB treatment [169]. Furthermore, in the context of H3122 lung adenocarcinoma, research findings suggested that the combination of ALK inhibition with a PI3K inhibitor (ZSTK474) or anticoccidial agent (salinomycin) reversed stem-like cell characteristics and impeded the development of acquired resistance. These findings underscore the significance of concurrent therapies for achieving optimal therapeutic effectiveness [170]. Moreover, a potential strategy for overcoming resistance involves co-targeting ROS1 and MEK with selumetinib in ROS1-rearranged NSCLC. Resistance is inevitable in ROS1-rearranged NSCLC, leading HCC78 cells to acquire KRAS G12C, amplify KRAS and FGF3, and sustain ERK activation [171].

Alternatively, simultaneous targeting of ALK and pan-ERBB-TKI substantially impeded colony formation and sustained downregulated pAKT levels for 72 h. HER3 knockdown induced varied effects in ALK+ cell lines, including reduced ALK expression and observable morphological changes. These findings imply HER3’s potential contribution to TKI resistance in ALK+ NSCLC, prompting further exploration of the combined targeting of ALK and HER3 in this context [172]. Moreover, different clinical ALK inhibitors exhibit varying impacts on cell migration, with MAP/microtubule affinity-regulating kinase (MARK) 2 and MARK3 demonstrating superior effects for brigatinib. The scientists pinpointed MARK2 and MARK3 as pertinent kinase targets for brigatinib in EML4-ALK+ NSCLC cells. Functional validation confirmed that inhibiting MARK2/3 through pharmaceutical means or genetic methods impedes cell migration. Brigatinib treatment induces inhibitory YAP1 phosphorylation downstream of MARK2/3. The results imply that brigatinib’s distinctive multitargeted activity, involving co-targeting MARK2/3, contributes to its heightened efficacy in preventing NSCLC cell migration compared to other ALK inhibitors [173].

In conclusion, the continuous evolution of ALK inhibitors, the advent of personalized medicine, and innovative co-targeting approaches promise extended PFS in NSCLC. Understanding resistance mechanisms, adopting liquid biopsies, and exploring diverse treatment strategies mark crucial steps toward managing ALK+ lung cancer as a chronic condition.

### 6.4. Other ALK-Innovative Approaches

In ALK+ cancer models, DNA vaccines directed against the ALK gene exhibited notable effectiveness [174]. These vaccines prompted specific immune reactions against ALK, fostering CD8+ T cell-mediated cytotoxicity and provoking IFN-γ responses. When combined with chemotherapy, ALK-DNA vaccination significantly extended the survival of mice afflicted with ALK+ lymphomas. In the context of ALK+ NSCLC models, the ALK-DNA vaccine triggered robust immune responses, curtailing tumor growth and elongating survival. However, lung tumors with ALK rearrangements establish an immunosuppressive setting, diminishing the efficacy of the ALK vaccine by upregulating PD-L1 expression. However, administering anti-PD-1 immunotherapy reinstates the vaccine’s effectiveness, implying that combining ALK vaccines with TKIs and ICIs could present a robust treatment strategy for ALK-driven NSCLC [175].

ALK vaccine pairing with ALK-TKIs notably delayed tumor relapse post-TKI treatment. Further research explored the treatment of ALK-rearranged NSCLC with ALK-TKI and ICIs. The findings revealed that ICIs were ineffective in prompting the rejection of ALK+ lung tumors. However, a vaccination with a single ALK peptide successfully reinstated the activation of ALK-specific CD8+ T cells. When coupled with ALK-TKIs, this vaccination eradicated lung tumors and impeded metastatic spread to the brain. The research additionally pinpointed human ALK peptides suitable for vaccination, demonstrating their immunogenicity in mice and their recognition by CD8+ T cells in individuals with NSCLC. This breakthrough implies the potential development of a clinical vaccine for treating ALK+ NSCLC [176].

Additionally, alternative ALK vaccines utilizing peptides or lipid vesicles encapsulating ALK antigens showcased potential in restraining tumor advancement in preclinical models. Further, the application of anti-EGF-VacAbs targeting EGF in ALK+ NSCLC cell lines amplified the effectiveness of ALK-TKIs, impeding the emergence of resistance and intercepting downstream oncogenic pathways [177]. These experimental findings propose an encouraging strategy for managing ALK-driven tumors, hinting at the possibility of ALK vaccines entering clinical trials.

On the other hand, scientists developed a highly sensitive NanoBiT LATS bioluminescent biosensor (BS) to track LATS kinase activity in the Hippo signaling pathway in lab settings and living organisms. This new biosensor showed greater sensitivity and stability than previous versions, even when expressed at significantly lower levels. Using this advanced biosensor, they could monitor LATS activity in live cells at physiologically relevant levels and simplify kinase activity analysis in vitro. Moreover, their study revealed an unprecedented interaction between ALK and the Hippo pathway, identifying a new mechanism involving the YAP/TAZ-PD-L1 axis. Targeting YAP/TAZ alone or in combination with ALK could offer a promising strategy for more effective treatment of cancers involving ALK or facing resistance to ALK inhibitors [178].

Alternatively, researchers recently investigated the role of the Nuclear Interaction Partner of ALK (NIPA) in a specific type of lymphoma induced by the NPM-ALK gene. Previous studies highlighted NIPA’s significance in cell division control and bone marrow failure but had yet to explore its involvement in NPM-ALK-driven lymphomas [179]. Researchers demonstrated that NIPA interacts with NPM-ALK, and its absence or downregulation led to significant impairment in the growth and transformation of cells associated with this lymphoma in lab tests. Further experiments in mice confirmed that removing or reducing NIPA in cells related to NPM-ALK-driven tumors prolonged survival without altering the tumors’ characteristics. Interestingly, the absence of NIPA affected a specific subpopulation of cells within the lymphoma, possibly impacting the disease’s onset and progression. These findings suggest that NIPA plays a crucial role in initiating this type of lymphoma, highlighting its potential as a target for future therapeutic interventions in this disease. Further investigations into the specific mechanisms of NIPA’s interaction with NPM-ALK and its role in tumor development could offer valuable insights for potential treatment strategies [180].

Furthermore, T-LAK cell-oriented protein kinase (TOPK), recognized as a potential therapeutic target in cancer, has been scrutinized in ALK+ NSCLC. The study identified ALK as an upstream kinase of TOPK, phosphorylating it specifically at Y74. This phosphorylation notably promotes tumor growth in ALK+ lung cancer cells, a finding supported by a phosphoproteomic analysis delineating downstream pathway involvement [181]. Comparatively, TOPK emerges as a superior target for cancer therapy compared to other direct downstream molecules of ALK, including Smad4, STAT3, PI3K, and PLC-γ. Clinical studies have consistently associated TOPK with a marker of poor prognosis in various cancers and an independent predictor for OS [182,183,184,185]. Encouragingly, inhibitors such as HI-032 and SKLB-C05, which target TOPK, have demonstrated promising potential.

Moreover, combining TOPK inhibition with alectinib, an ALK inhibitor, has shown remarkable synergy in impeding cell proliferation and promoting apoptosis. This combined approach proposes a promising strategy to counter drug resistance in ALK+ NSCLC [181]. These research findings advance our understanding of ALK’s oncogenic signaling network and suggest the potential efficacy of co-inhibition of ALK and TOPK as a novel therapeutic strategy to treat ALK+ NSCLC and potentially delay the onset of drug resistance.

## 7. Concluding Remarks

ALK+ NSCLC, affecting about 5% of cases, is characterized by a mutation in the ALK gene, leading to poor life expectancy and a high risk of brain metastases. Unmet needs in metastatic NSCLC include the development of treatments that improve survival, reduce toxicity, and effectively address brain metastases. Evaluating FDA-approved ALK inhibitors in advanced NSCLC highlights their unique effectiveness and safety profiles. Crizotinib exhibits notable benefits regarding PFS and ORR; however, multiple studies consistently position alectinib as the superior option. Alectinib distinguishes itself with extended PFS, increased CNS activity, and excellent patient-reported outcomes.

In first-line scenarios, clinical trials featuring alectinib, brigatinib, and lorlatinib demonstrated higher CNS response rates (86–94%, 67–78%, and 42–82%, respectively) compared to crizotinib (16–71%). Median PFS varied, with crizotinib (5.6–7.4 months) exhibiting lower rates than alectinib (not reached), brigatinib (24.0 months), and ceritinib (10.7–25.2 months). Next-generation TKIs are preferred for patients with advancing brain metastases. Despite their notable efficacy, obstacles persist in accessing personalized therapy, particularly concerning next-generation ALK-TKIs in patients progressing on crizotinib [186].

In contrast, while brigatinib effectively extends PFS, it has a higher incidence of adverse effects than crizotinib. Moreover, ceritinib’s ability to enhance PFS and reduce disease progression or mortality emphasizes its clinical significance. As a primary treatment, lorlatinib displays promising outcomes by demonstrating superior PFS and more patients without disease progression at 12 months than crizotinib. However, it is essential to note that these treatments have varying impacts on patients’ quality of life due to their distinct adverse effect profiles. These findings provide essential insights for tailoring treatments based on individual patients’ needs and tolerability, thereby shaping personalized strategies for managing ALK+ NSCLC. Additionally, exploring interactions between ALK and immunotherapy and innovative methods such as ALK vaccines, biosensors, and targeted pathway approaches offers potential avenues for future interventions in ALK-driven cancers.

However, the estimated cost of ALK-targeted therapy, such as lorlatinib, is approximately USD 8982 per 28-day cycle [187]. While lorlatinib demonstrated superior PFS compared to brigatinib in the overall patient population, there was no significant difference observed between the two in the subgroup of patients with CNS metastases [188]. The Canadian Agency for Drugs and Technologies in Health (CADTH) recommended reimbursement for patients without prior systemic treatment for advanced NSCLC. CADTH advises using lorlatinib as a standalone drug with a cost not exceeding that of alectinib or brigatinib [187]. This guidance from CADTH suggests carefully considering cost-effectiveness and aims to ensure patients can access effective treatments while managing healthcare expenditures. The specific cost comparison to other therapies underscores the importance of evaluating the economic value of medications and their clinical efficacy when making recommendations for reimbursement in healthcare systems. This approach helps optimize resource allocation and ensure equitable access to treatments for eligible patients.

## Figures and Tables

**Figure 1 biomedicines-12-00297-f001:**
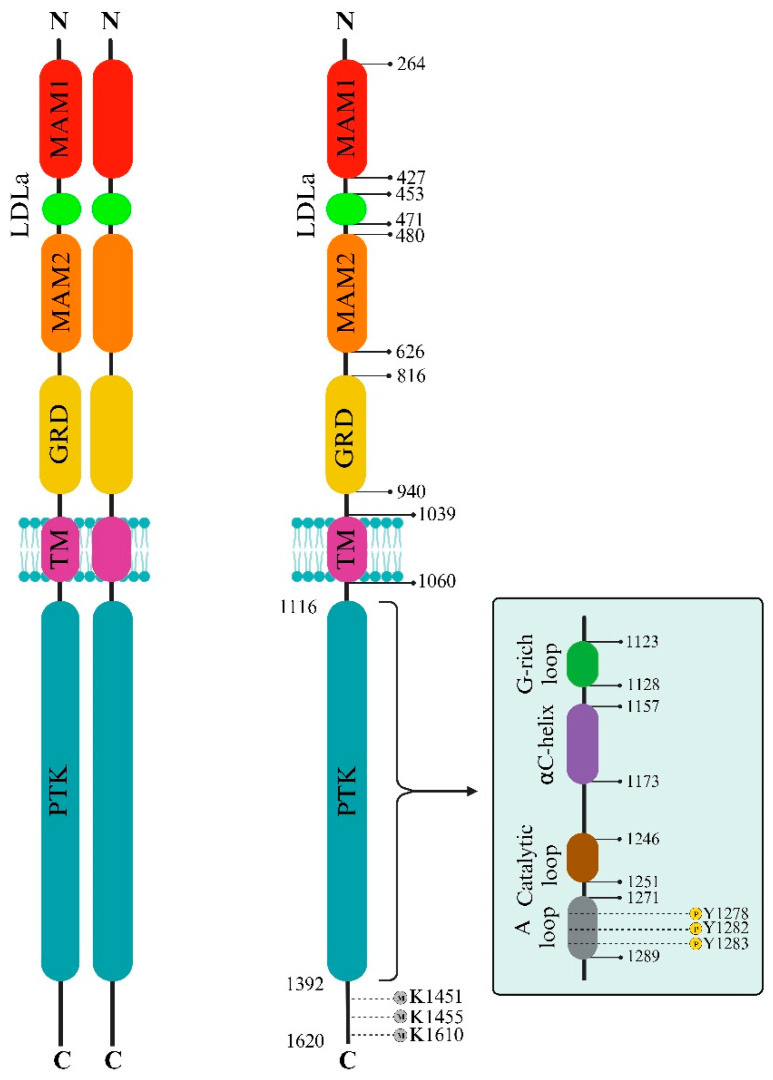
ALK Domain Structure and Regulatory Elements. This comprehensive depiction illustrates ALK’s extracellular and intracellular aspects, emphasizing key structural domains and functional motifs. The extracellular domain comprises sections crucial for ligand binding and potential activation, including MAMs and LDLa domains. Within the intracellular protein kinase domain (PTK), essential regulatory segments governing ALK’s active and inactive states are highlighted, shedding light on its allosteric control and potential therapeutic targeting. Notably, the A-loop, housing pivotal amino acid residues Y1278, Y1282, and Y1283, drives ALK activation and downstream signaling. In contrast, specific C-terminal lysine residues serve as targets for methylation, contributing to regulatory functions. Furthermore, this figure delineates the duality of ALK functionality: ligand-dependent dimerization (**Left**) and ligand-independent monomeric activity (**Right**).

**Figure 2 biomedicines-12-00297-f002:**
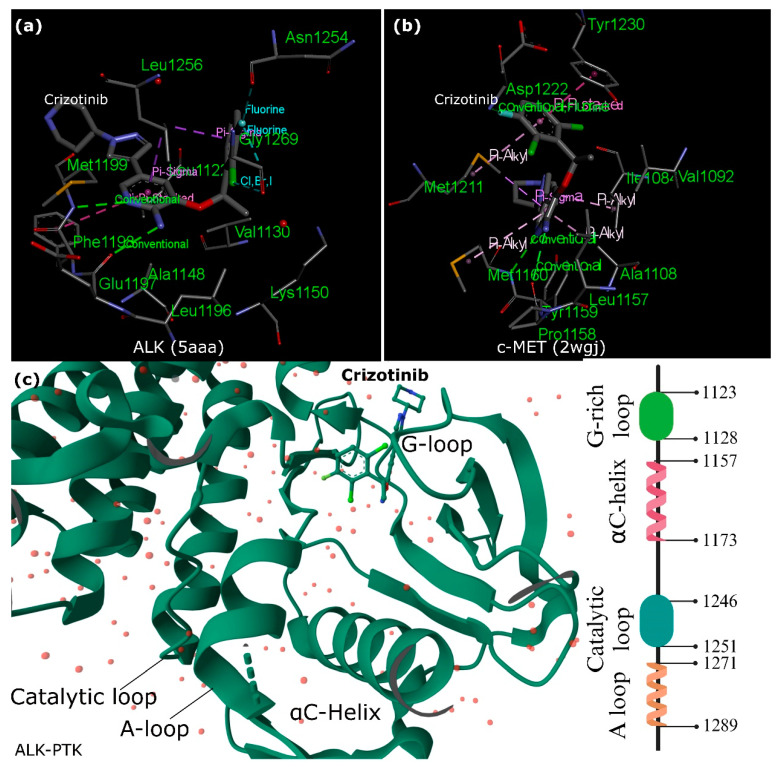
Crizotinib Interactions with ALK and c-MET. (**a**,**b**) Comparison of Crizotinib’s Interactions with (**a**) ALK (PDB id: 5aaa [40]) and (**b**) c-MET (PDB id: 2wgj [33]). The figure illustrates distinct binding sites of Crizotinib within unphosphorylated c-MET and ALK-PTK, highlighting crucial π interactions and conventional hydrogen bonds. Fundamental interactions, such as those with specific amino acids (e.g., M1211 in c-MET), contribute significantly to Crizotinib’s inhibitory effect. (**c**) ALK-PTK (PDB id: 5aaa). The figure details the binding interactions of Crizotinib within the ALK-PTK domain, emphasizing critical regions such as the G-loop, A-loop, and catalytic loop. ALK’s activation loop (A-loop) with the conserved Asp-Phe-Gly (DFG) sequence plays a pivotal role in regulating ALK’s active and inactive states. Hydrophobic spines named the “regulatory” and “catalytic” contribute to vital allosteric control between lobes, impacting the transition between active and inactive states. This figure was designed using the BIOVIA Discovery Studio Visualizer (v.21.1).

**Figure 3 biomedicines-12-00297-f003:**
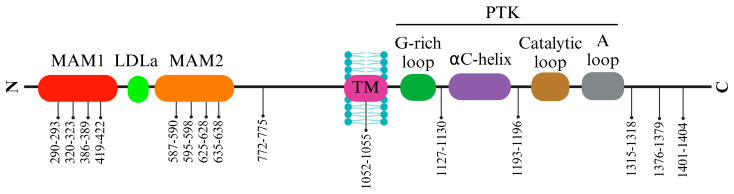
ALK Sequences as LC3-Interacting Targets. LC3, a mammalian counterpart to yeast Atg8, is a precise marker for autophagy monitoring. The diagram highlights various LC3-interacting region (LIR) motifs across ALK’s diverse domains, indicating a direct link between ALK and autophagy. This connection suggests a complex interplay between suppressing ALK kinase activity and activating autophagy, potentially complicating therapeutic approaches for NSCLC and similar conditions. Understanding autophagy’s dual role in cancer, both as an immune response facilitator and a promoter of tumor growth, underscores the need to classify ALK+ NSCLC based on HGF/c-MET signaling or autophagy-related subtypes. This stratification enables customized treatment strategies for optimizing patient outcomes.

**Figure 4 biomedicines-12-00297-f004:**
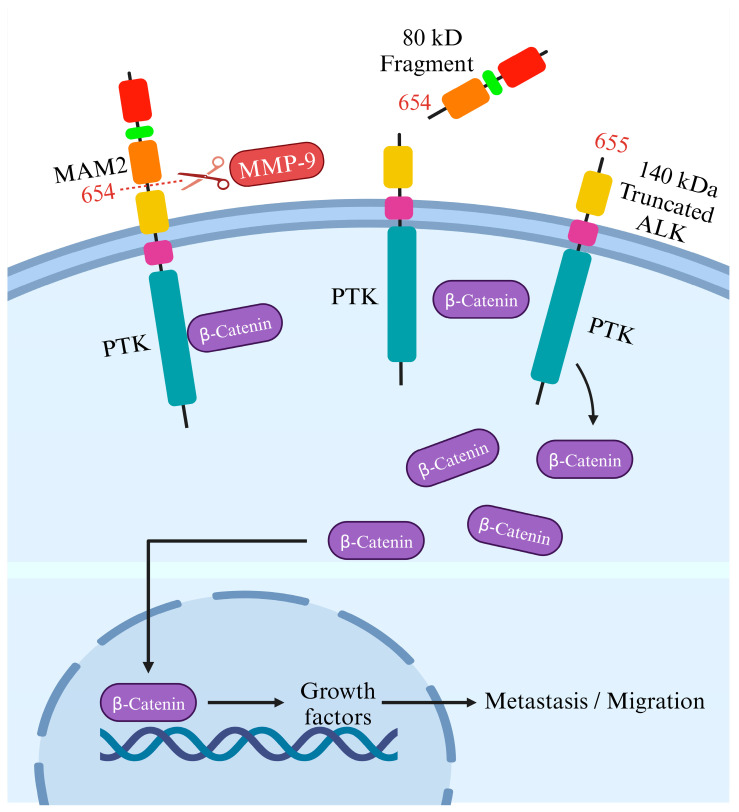
ALK Cleavage and its Impact on Tumor Cell Behavior. This illustration demonstrates the complex process of ALK cleavage in the tumor microenvironment by MMP9, resulting in the release of an 80 kDa extracellular fragment and a 140 kDa truncated ALK at the membrane. The cleavage of ALK by MMP9 in the tumor microenvironment has diverse effects on immune cell activity and tumor cell migration. While intracellular domain cleavage may aid ALK-targeted therapy, extracellular cleavage fosters ALK-related tumor formation and cellular movement, particularly in neuroblastoma.

**Figure 5 biomedicines-12-00297-f005:**
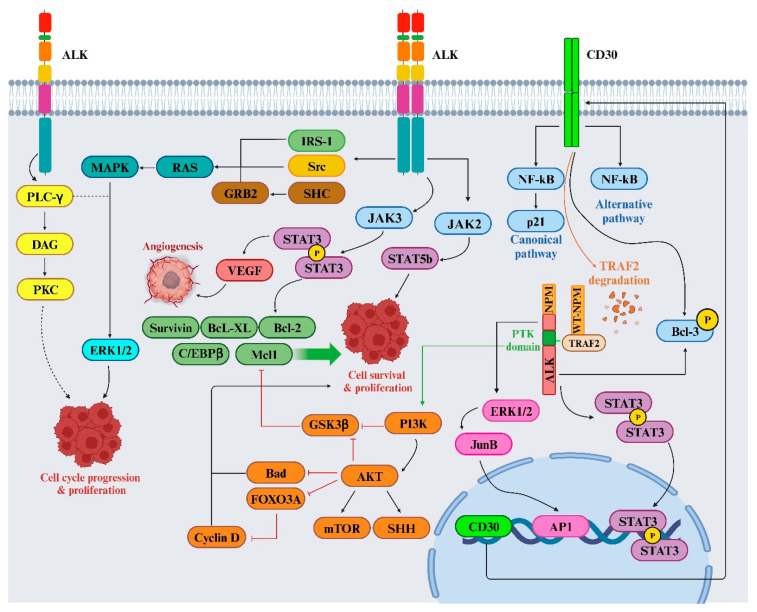
ALK Signaling Pathways and Therapeutic Implications. This comprehensive visualization outlines the intricate network of ALK signaling pathways, depicting the influence of both membrane wild-type ALK and cytoplasmic NPM-ALK on Akt, MAPK, and STAT3 cascades. ALK mutations and fusion proteins drive varied signaling pathways linked to cell transformation, survival, and therapeutic resistance in ALK+ cancers, notably NSCLC. Despite clinical use of ALK inhibitors (crizotinib, alectinib, ceritinib), resistance emerges due to diverse ALK mutations and fusion proteins, activating multiple adaptors and signaling cascades. These alterations activate mitogenic signaling (RAS/MAP kinase), PI3K, and PLC-γ pathways, impacting anti-apoptotic signaling, cell cycle regulation, and cellular metabolism. Strategies targeting ALK, MYC, Src, JNK/c-Jun, and Wnt/β-catenin pathways show potential against ALK resistance. Novel inhibitors (ZX-29, XMU-MP-5) demonstrate efficacy against diverse ALK mutations. Moreover, the co-expression of CD30, a TNFR superfamily member, alongside ALK in certain cancers—especially ALCL—establishes a distinct ALK+, CD30+ immunophenotype. This co-expression aids ALCL diagnosis and guides targeted therapies, underscoring the diagnostic and therapeutic significance of ALK-CD30 co-expression patterns. Abbreviations: ALK: Anaplastic Lymphoma Kinase; AP1: Activator Protein-1; Bcl-2: B-cell lymphoma 2; DAG: Diacylglycerol; ERK: Extracellular Signal-Regulated Kinase; GRB2: Growth Factor Receptor-Bound Protein 2; IRS: Insulin Receptor Substrate; JAK: Janus Kinase; MAPK: Mitogen-Activated Protein Kinase; Mcl1: Myeloid cell leukemia 1; mTOR: Mammalian Target of Rapamycin; NPM: Nucleophosmin; NF-κB: Nuclear Factor-kappa B; PI3K: Phosphoinositide 3-Kinase; PKC: Protein Kinase C; PLC-γ: Phospholipase C-gamma; PTK: Protein Tyrosine Kinase; RAS: Rat Sarcoma; SHC: SHC-transforming protein; SHH: Sonic Hedgehog; Src: Proto-oncogene tyrosine-protein kinase Src; STAT: Signal Transducer and Activator of Transcription; TRAF2: Tumor Necrosis Factor Receptor-Associated Factor 2; VEGF: Vascular Endothelial Growth Factor.

**Figure 6 biomedicines-12-00297-f006:**
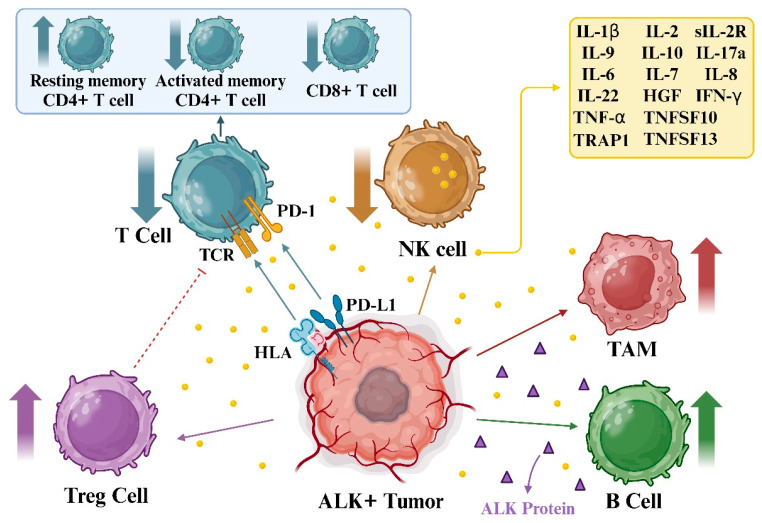
Interaction between ALK+ Tumors and Immune Cells. The complex landscape of ALK gene alterations significantly contributes to the diverse spectrum of human cancers, revolutionizing therapeutic approaches. These alterations influence the immune milieu within tumors, prompting the exploration of immunotherapy as a promising clinical avenue. Interleukins (IL-6, IL-8, IL-10) are implicated in NSCLC progression, notably in ALK+ patients, affecting the immune landscape via Toll-like receptors (TLRs) and interleukin interactions. Serum markers such as soluble IL-2R (sIL-2R) and CD30 provide insights into disease activity and immune response in ALCL, particularly ALK+ patients, impacting prognosis and treatment assessment. Cytokine profiles in ALCL patients reveal distinct patterns correlating with disease stages, reflecting tumor characteristics and immune responses. Moreover, cytokine levels such as IL-6, IL-8, and IL-10 serve as potential indicators for disease progression in ALK+ NSCLC, offering additional monitoring options. Abbreviations: HGF: Hepatocyte Growth Factor; HLA: Human Leukocyte Antigen; IFN-γ: Interferon-gamma; IL: Interleukin; sIL-2R: Soluble Interleukin-2 Receptor; TCR: T-cell Receptor; TAM: Tumor-Associated Macrophage; TNF-α: Tumor Necrosis Factor-alpha; TNFSF10: Tumor Necrosis Factor Superfamily Member 10; TRAP1: TNF Receptor-Associated Protein 1.

**Figure 7 biomedicines-12-00297-f007:**
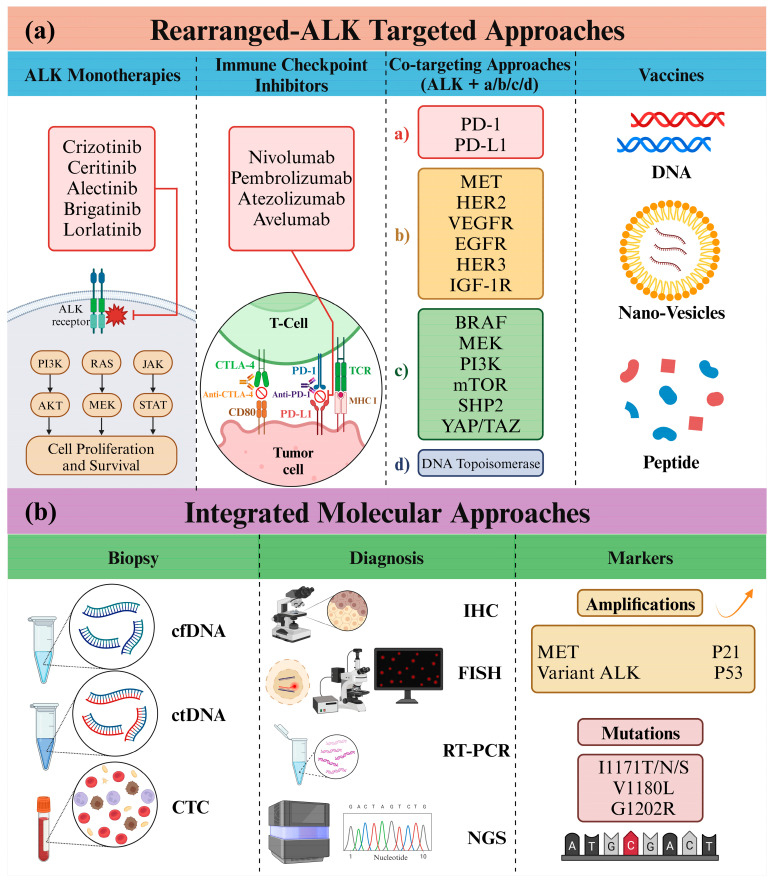
Schematic depiction of therapeutic strategies targeting rearranged ALK. (**a**) Rearranged-ALK Targeted Approaches. This section outlines ALK monotherapy with Crizotinib, Ceritinib, Alectinib, Brigatinib, and Lorlatinib. It also highlights immune checkpoint inhibitors Nivolumab, Pembrolizumab, Atezolizumab, and Avelumab, along with co-targeting approaches involving ALK combined with a/b/c/d, where (a–d) denote distinct targetable signaling proteins. The co-targeting approach involves the inhibition of various proteins, including Programmed Cell Death Protein 1 (PD-1), Programmed Death-Ligand 1 (PD-L1), Mesenchymal Epithelial Transition Factor (MET), Human Epidermal Growth Factor Receptor 2 (HER2), Vascular Endothelial Growth Factor Receptor (VEGFR), Epidermal Growth Factor Receptor (EGFR), HER3, Insulin-Like Growth Factor 1 Receptor (IGF-1R), B-Raf Proto-Oncogene (BRAF), Mitogen-Activated Protein Kinase (MEK), Phosphatidylinositol 3-Kinase (PI3K), Mammalian Target of Rapamycin (mTOR), Src Homology 2 Domain-Containing Phosphatase 2 (SHP2), Yes-Associated Protein/Transcriptional Coactivator with PDZ-Binding Motif (YAP/TAZ), DNA Topoisomerase, and vaccination modalities (DNA, Nano-Vesicles, Peptide). (**b**) Integrated Molecular Approaches. Biopsy specimens encompass circulating tumor DNA (ctDNA), cell-free DNA (cfDNA), and circulating tumor cells (CTC). Diagnostic techniques include Next-Generation Sequencing (NGS), Reverse Transcription-Polymerase Chain Reaction (RT-PCR), Immunohistochemistry (IHC), and Fluorescence In Situ Hybridization (FISH). Key markers involve amplifications, such as MET, P21, Variant ALK, and P53, and mutations such as I1171T/N/S, V1180L, and G1202R.

## Data Availability

Not applicable.

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
