# Peer review of "Unraveling the Potential of ALK-Targeted Therapies in Non-Small Cell Lung Cancer: Comprehensive Insights and Future Directions"

_biomedicines, 2024, doi:10.3390/biomedicines12020297_

Round 1
Reviewer 1 Report
Comments and Suggestions for Authors
The authors have thoroughly examined non-small cell lung cancer (NSCLC) with anaplastic lymphoma kinase (ALK) rearrangements, exploring the mechanisms and recent information related to both ALK-targeted therapy and immunotherapy.
In general, the review is well-written and comprehensively presented. In the past three years, there have been more than 50 reviews regarding ALK in NCSLC. The authors should clearly emphasize the novelty of their review and what it adds to the field.
Some sections are brief, particularly those covering recent advances in ALK-targeted therapies. The authors should dwell more about these areas and future role of next-generation sequencing and molecular diagnostics, such as liquid biopsies, in ALK-targeted therapy and drug resistance. The role of co-targeting approaches to reduce resistance and disease relapse should be presented.
Under section 4, the authors are advised to add a figure summarizing future directions in ALK-targeted approaches in NCSLC.
Abbreviations are well covered but need to be introduced in the figure legends.
The authors can reference few relevant recent reviews related to their topic namely:
Zia et al. Transl Lung Cancer Res. 2023
Desai and Lovly. Transl Lung Cancer Res. 2023
Fukuda and Yoshida. Expert Rev Anticancer Ther. 2023
Meador and Piotrowska. Transl Lung Cancer Res. 2023
Minor edits:
Use in title non-small cell lung cancer instead of NSCLC.
Replace the second “delves” in the abstract by explores.
Author Response
Dear Reviewer,
I would like to express my gratitude for your thoughtful and constructive review of our manuscript titled " Unravelling the Potential of ALK–Targeted Therapies in Non-Small Cell Lung Cancer: Comprehensive Insights and Future Directions". Your feedback has been instrumental in improving the quality and clarity of our work, and we appreciate the time and effort you dedicated to this review.
We have attached a tracked document ('track-revised.docx'), providing a transparent view of all changes made during the revision process. We firmly believe that these modifications have significantly strengthened the manuscript. In addition, point-to-point rebuttals have been attached at the end of this letter. We are confident that the revised manuscript warrants further consideration for publication in Biomedicines.
We look forward to a rapid editorial decision.
With my best regards,
Parham Jabbarzadeh Kaboli, Ph.D.

Reviewer 2 Report
Comments and Suggestions for Authors
please highlight the research method to select the studies throughout the literature by understanding the characteristics, purposes and results. Place more emphasis on efficacy and tolerability.The relationship between the immune system and ALK rearrangement should be better explained. Please point out the side effects of combined nivolumab-crizotinib.
"Lung cancer comprises NSCLC (81% of cases) and small cell lung cancer (SCLC) (14% of cases). In the U.S., NSCLC dominates and is projected to affect around 238,340 adults by 2023, resulting in 127,070 deaths. Globally, 2,206,771 people were diagnosed with lung cancer in 2020, encompassing both NSCLC and SCLC cases" I would put this piece at the beginning of the introduction
I also suggest discussing the methods most commonly used in clinical practice to detect ALK translocation, including immunohistochemistry and circulating tumor cells detection.
In this regard please include the following reference for discussion
Transl Lung Cancer Res. 2021 Jan;10(1):80-92.
Author Response

(The authors gave the same response as above.)

Round 2
Reviewer 1 Report
Comments and Suggestions for Authors
The authors need to revise their search strategy and use non-small cell lung cancer in title and ALK in title as well as reviews with NSCLC in title and ALK in title. They are going to capture most of the manuscripts related to this topic. When they use NSCLC in title and ALK in title, they will only capture few articles. Indeed, there have been more than 50 reviews in the past three years regarding ALK in NCSLC. They need to do this search to include relevant manuscripts and complete their search strategy.
Regarding all the other comments, the authors have satisfactorily addressed my concerns and made the necessary revisions.
Author Response
Dear Reviewer,
Thank you for providing constructive feedback on our manuscript. We appreciate your thorough review and are glad to learn that the revisions addressing your other concerns have been satisfactory.
We understand the importance of refining our search strategy to ensure a comprehensive literature review. In response to your suggestion, we have conducted an extensive search using the specified method, including "non-small cell lung cancer" and "ALK" in the title, as well as reviews with "NSCLC" and "ALK" in the title. This enhanced search approach has resulted in the identification of 25 additional references. We also updated the search methodology in the abstract.
We have diligently updated the manuscript to incorporate the relevant findings from these additional sources, and 25 references were added. We believe that this expanded search will contribute to the depth and completeness of our manuscript, capturing a broader range of literature on the topic.
Your guidance has been invaluable, and we appreciate your commitment to ensuring the quality of our work. Please let us know if you have any further suggestions or specific areas for improvement. We are dedicated to delivering a comprehensive and well-informed manuscript.
Once again, thank you for your time and valuable input.
Sincerely,
Parham Jabbarzadeh Kaboli, Ph.D.
GIBS, China Medical University.